# The Rad53CHK1/CHK2-Spt21NPAT and Tel1ATM axes couple glucose tolerance to histone dosage and subtelomeric silencing

Christopher Bruhn [1 ✉], Arta Ajazi[1], Elisa Ferrari[1], Michael Charles Lanz[2], Renaud Batrin[3], Ramveer Choudhary[1], Adhish Walvekar[4], Sunil Laxman [4], Maria Pia Longhese [5], Emmanuelle Fabre [3], Marcus Bustamente Smolka [2] & Marco Foiani [1,6 ✉]

The DNA damage response (DDR) coordinates DNA metabolism with nuclear and non-nuclear processes. The DDR kinase Rad53CHK1/CHK2 controls histone degradation to assist DNA repair. However, Rad53 deficiency causes histone-dependent growth defects in the absence of DNA damage, pointing out unknown physiological functions of the Rad53-histone axis. Here we show that histone dosage control by Rad53 ensures metabolic homeostasis. Under physiological conditions, Rad53 regulates histone levels through inhibitory phosphorylation of the transcription factor Spt21NPAT on Ser276. Rad53-Spt21 mutants display severe glucose dependence, caused by excess histones through two separable mechanisms: dampening of acetyl-coenzyme A-dependent carbon metabolism through histone hyperacetylation, and Sirtuin-mediated silencing of starvation-induced subtelomeric domains. We further demonstrate that repression of subtelomere silencing by physiological Tel1ATM and Rpd3HDAC activities coveys tolerance to glucose restriction. Our findings identify DDR mutations, histone imbalances and aberrant subtelomeric chromatin as interconnected causes of glucose dependence, implying that DDR kinases coordinate metabolism and epigenetic changes.

[1] The FIRC Institute of Molecular Oncology (IFOM), Via Adamello 16, 20139 Milan, Italy. [2] Department of Molecular Biology and Genetics, Weill Institute for Cell and Molecular Biology, Cornell University, Ithaca, NY 14853, USA. [3] Université de Paris, Laboratoire Génomes, Biologie Cellulaire et Thérapeutiques, CNRS UMR7212, INSERM U944, Centre de Recherche St Louis, F-75010 Paris, France. [4] Institute for Stem Cell Science and Regenerative Medicine (inStem), Bangalore, Karnataka 560065, India. [5] Dipartimento di Biotecnologie e Bioscienze, Università degli Studi di Milano-Bicocca, Edificio U3, Piazza della Scienza 2, 20126 Milan, Italy. [6] Università degli Studi di Milano, Via Festa del Perdono 7, 20122 Milan, Italy. ✉email: christopher.bruhn@ifom.eu; marco.foiani@ifom.eu

Glucose is the primary carbon source for many tissues, cancer cells and unicellular organisms to fuel energy production and biosynthetic reactions. The glucose demand depends on energy consumption and anabolic activity; a prominent example is the glucose requirement of rapidly dividing cancer cells that undergo metabolic reprogramming[1]. Targeting alterations in glucose metabolism[2] and nutrient restriction[3] are promising approaches to complement classical cancer treatment.

Cell division is regulated by the metabolic state through the readout of energy charge and metabolite pools[4]. This sensing is implemented by the action of key metabolites on signaling by either stimulating enzyme activities or acting as rate-limiting post-translational modification donors. Acetyl-coenzyme A (Ac-CoA) is a central carbon metabolite and the acetyl donor in all acetylation reactions[5,6]. Although mitochondrial Ac-CoA fuels oxidative phosphorylation and thus represents a major energy source, a separate nucleocytoplasmic pool drives essential biosynthetic reactions such as the synthesis of lipids. Ac-CoA is rate-limiting for the acetylation of histones from yeast to human[7–10]. Together with the relative activity of histone acetyltransferases (HATs) and histone deacetylases (HDACs), Ac-CoA levels influence histone acetylation, chromatin state, and gene expression, coupling metabolic changes to cell cycle control[11–13]. Although this sensing mechanism likely relies on a physiological balance of Ac-CoA vs Ac-CoA acceptors, it is not known whether and to what extent Ac-CoA metabolism is affected by variations in histones as Ac-CoA acceptors.

Budding yeast has the capacity to adapt to various environments, and the underlying sensing processes are evolutionarily conserved[14]. High glucose supply supports fermentative metabolism, which is similar to the aerobic glycolysis common to tumor cells. In contrast, glucose restriction leads to the de-repression of starvation response genes and causes a switch to respiratory metabolism[15]. This regulation involves the coordinate de-repression of functional silent gene clusters, which are mainly located within subtelomeric regions[16]. The major player mediating the silencing of these clusters is the Sir complex, which propagates gene silencing from telomeres through histone de-acetylation, catalyzed by the HDAC Sir2[SIRT1]. Regulators of histone acetylation[17] and telomere length[18] can influence silent chromatin propagation, but the consequent impact on metabolic adaptation is not known.

The DNA damage response (DDR) is a conserved signaling network that safeguards genome stability. It is also a crucial coordinator of dNTP[19] and oxidative[20–22] metabolism throughout the cell cycle. In budding yeast, the apical DDR kinases Mec1[ATR] and Tel1[ATM] and their downstream kinase Rad53[CHK1/CHK2] mediate most DDR functions. We have previously identified a role of the glucose-regulated phosphatase PP2A in the attenuation of DDR activity, implying a coordination of DDR and glucose metabolism[23]. Consistently, Rad53 is required for growth in non-glycolytic carbon sources[24]. However, none of the currently established DDR functions can mechanistically explain this phenomenon, suggesting an unknown function of the DDR in regulating the tolerance to glucose restriction.

In the current study, we show that Rad53 mediates tolerance to glucose restriction through inhibitory phosphorylation of the conserved histone transcription factor (TF) Spt21[NPAT]. Excess histones accumulating in rad53Δ mutants confer sensitivity to glucose restriction by affecting Ac-CoA-dependent metabolism and enhancing sirtuin-dependent gene repression. Tel1 contributes to glucose tolerance by modulating subtelomeric silencing. Our data thus identify independent roles of Rad53-Spt21 and Tel1 axes in modulating glucose dependence.

## Results

### A Rad53-histone axis mediates glucose restriction tolerance.

We have previously found that the glucose-regulated phosphatase PP2A attenuates DDR activity[23]. We hypothesized that DDR activity may also affect glucose metabolism. We therefore tested by semi-quantitative spot assay whether the DDR kinases Mec1[ATR], Tel1[ATM], Rad53[CHK1/CHK2], Chk1[CHK1], and Dun1 were required to survive under glucose limitation. mec1Δ and rad53Δ mutants can be kept alive by deleting the SML1 gene, encoding the ribonucleotide reductase (RNR) inhibitor[19]. We found that sml1Δrad53Δ mutants were specifically sensitive to glucose restriction (Fig. 1a and Supplementary Fig. 1a; D = dextrose/glucose, DR = glucose restriction below 0.05%). We confirmed this observation in rad53-K227A hypomorphic mutants that retain 10% of the kinase activity[25] (Fig. 1a). Rad53 is activated by Mec1 and Tel1[26]. Ablation of MEC1 or TEL1 was not sufficient to sensitize to glucose restriction; mec1Δtel1Δ mutants exhibited a near-lethal phenotype already in high-glucose conditions (Fig. 1a). Rad53 activates the Dun1 kinase to promote dNTP synthesis, through inhibition of Sml1. However, dun1Δ mutants were not glucose-dependent, and deletion of SML1 did not alleviate the glucose dependence of the rad53-K227A mutant (Fig. 1a), suggesting independence from RNR regulation. Chk1 is a DDR protein kinase acting downstream of Tel1. chk1Δ mutants did not exhibit sensitivity to low glucose (Fig. 1a).

Rad53 promotes the degradation of excess histones, and loss of this regulation inhibits the proliferation of sml1Δrad53Δ mutants[27,28]. We asked if aberrant histone regulation also caused glucose dependence. Indeed, lowering histone H3 and H4 levels by disrupting the major H3/H4 locus, HHT2-HHF2 (hht2Δ), increased the fitness of sml1Δrad53Δ under glucose restriction and ameliorated the near-lethal phenotype of mec1Δtel1Δ double mutants (Fig. 1b). We asked if lack of Rad53 activity caused histone accumulation under glucose restriction, and monitored histone levels after acute Rad53 depletion with the Auxin-inducible degron system (sml1Δ tet-Rad53-Myc-AID). Core histones H3 and H4 accumulated rapidly after Rad53 depletion (2 h) (Fig. 1c). We observed less histone accumulation in high-glucose conditions.

### Rad53 controls histone dosage through Spt21 inhibition.

We performed a quantitative phosphoproteomic screen for Rad53 targets controlling histone levels under glucose restriction. As the function of Rad53 in tolerating glucose restriction was likely independent of Dun1 signaling (Fig. 1a) and upstream of histone proteins, we used sml1Δrad53Δhht2Δ and sml1Δdun1Δhht2Δ as experimental and control strains, respectively (Fig. 1d and Supplementary Fig. 1b). Direct comparison of phosphopeptides from both strains allowed the specific identification of Spt21–S276 as Rad53-dependent, Dun1-independent phosphorylation site (sml1Δrad53Δhht2Δ < sml1Δdun1Δhht2Δ) (Fig. 1d, Supplementary Fig. 1c, d; Supplementary Table 1). Spt21 is the yeast homolog of human NPAT, and both act as major inducers of histone expression during S phase[29–31]. Spt21–S276 corresponds to NPAT-S981 and is located within a strong consensus motif for Rad53 phosphorylation (S/T-Ψ)[32] (Fig. 1d). We hypothesized that Rad53 inhibits Spt21 to control histone gene expression. Indeed, deletion of SPT21 completely rescued the accumulation of histone mRNA and proteins after acute Rad53 depletion (Fig. 1e, f, and Supplementary Fig. 1e). Similarly, deletion of its transcriptional activator partner, Spt10, prevented histone accumulation (Supplementary Fig. 1f). Consistent with the S phase-specific expression of Spt21, G1 arrest also prevented histone accumulation (Supplementary Fig. 1g). Importantly, deletion of

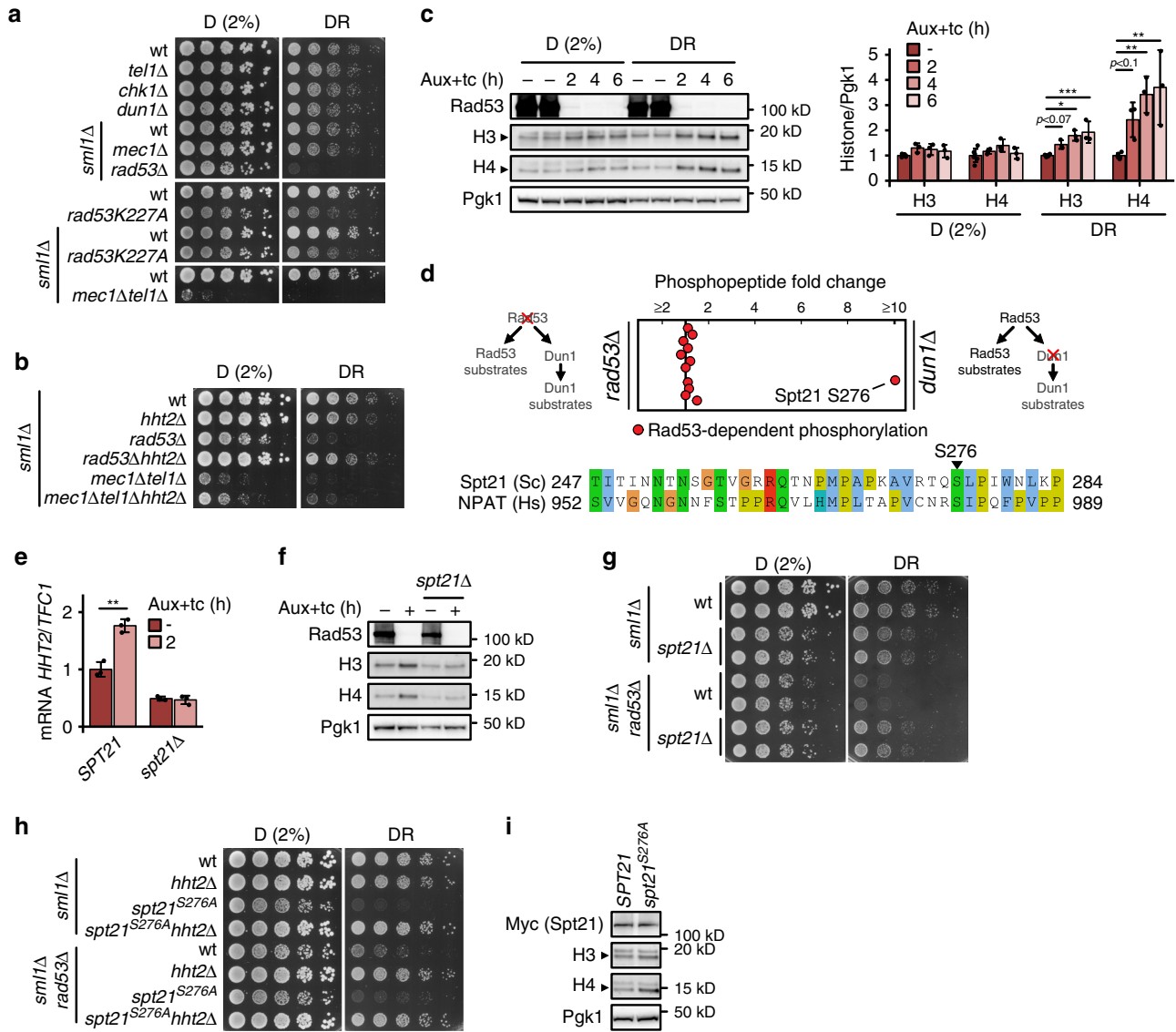

**Fig. 1 A Rad53-Spt21 axis mediates tolerance to glucose restriction. a**, **b** $10^7$ cells/mL were serially diluted (1:6), spotted on YP plates with the indicated carbon sources and grown for 2d. **c** Rad53-AID cells were adapted to normal or low glucose, Rad53 was depleted with Auxin and tetracycline, and samples were collected at the indicated times for western blot analysis ($n = 3$ independent experiments). The bar chart (right panel) provides a quantification of histone levels normalized by Pgk1. Significances were calculated with one-way ANOVA ($p_{DR\_H3} = 0.00027$, $p_{DR\_H4} = 0.0011$) with post hoc Tukey HSD test ($p_{DR\_H3\_2h\_vs\_UT} = 0.066$, $p_{DR\_H3\_4h\_vs\_UT} = 0.0015$, $p_{DR\_H3\_6h\_vs\_UT} = 0.00042$, $p_{DR\_H4\_2h\_vs\_UT} = 0.094$, $p_{DR\_H4\_2h\_vs\_UT} = 0.0044$, $p_{DR\_H4\_2h\_vs\_UT} = 0.0019$). **d** Top: quantitative mass spectrometry analysis of Dun1-dependent ($rad53\Delta$) and Rad53-dependent ($dun1\Delta$) phosphorylation events in low glucose. Each dot corresponds to a phosphorylation event previously found to exhibit Rad53 dependency (see Methods). The horizontal position represents the relative fold-change. ($n = 2$ independent label swap experiments). Bottom: alignment of budding yeast Spt21 and human NPAT around Spt21-S276. **e**, **f** Rad53-AID cells with the indicated *SPT21* genotype were adapted to low glucose and Rad53 was depleted with Auxin and tetracycline for 2h **e** or 4h **f**. Samples were collected to quantify histone mRNAs by RT-qPCR **e** or proteins by western blot analysis **f** ($n = 3$ independent replicate cultures in **e**, representative blot in **f**). Significance in **e** was calculated by Student's *t* test (two-sided, unpaired, $p_{SPT21\_2h\_vs\_UT} = 0.0017$). **g**, **h** $10^7$ cells/mL were serially diluted (1:6), spotted on YP plates with the indicated carbon sources and grown for 2d. **i** Spt21-AID and Spt21$^{S276A}$-AID cells were adapted to low glucose and Auxin for 16 h. Auxin was washed away to stabilize Spt21 and samples were collected after 6 h to quantify proteins by western blot analysis. Bars plots with error bars represent mean values and standard deviation, *D* glucose; *DR* glucose restriction (0.01% **a**, **b**, **g** or 0.04% **h** in solid media, 0.02% in liquid media), *Aux* Auxin; *tc* tetracycline. \**p* < 0.05, \*\**p* < 0.01, \*\*\**p* < 0.001.

*SPT21* also rescued the slow growth and glucose dependence of *sml1Δrad53Δ* mutants (Fig. 1g).

Spt21–S276 is not phosphorylated in the absence of Rad53. To mimic the constitutive un-phosphorylated state, we constructed the *spt21S276A* mutation in the endogenous *SPT21* locus. *spt21S276A* was sufficient to cause slow growth, glucose dependence and histone accumulation (Fig. 1h, i). Deletion of *RAD53*

did not further reduce growth, and deletion of *HHT2* fully rescued growth and glucose dependence of *sml1Δrad53Δspt21S276A* mutants (Fig. 1h), supporting a linear pathway of histone regulation by Rad53 and Spt21. In summary, we suggest that inhibition of Spt21 by basal Rad53 activity controls histone levels during an unperturbed S phase, and thereby mediates resistance to glucose restriction.

**Excess histones suppress a subtelomeric starvation response.**
We next asked how excess histones in Rad53–Spt21 axis mutants imposed glucose dependence. Glucose restriction de-represses starvation response genes that facilitate stress resistance and metabolic adaptation. We hypothesized that excess histones may interfere with this program. To measure the transcriptional response to glucose depletion, we profiled the transcriptomes of *sml1Δ*, *sml1Δrad53Δ*, and *sml1Δrad53Δhht2Δ* cells after an acute switch (1 h) or adaptation (20 h) to a non-glycolytic carbon source (ethanol), which relieves glucose repression similar to glucose withdrawal (Fig. 2a; Supplementary Data 1). We categorized switch-inducible genes as either transient (induced exclusively at 1 h) or adaptive (induced at 20 h) (Fig. 2a). We then compared the expression levels of all switch-inducible genes (Supplementary Data 1) in *sml1Δrad53Δ* and *sml1Δ* cells at 1 h after the switch and found that *RAD53* deletion did not affect their mean expression (Fig. 2b). Similarly, *RAD53* deletion had no effect in the average expression level of adaptively induced genes at 20 h after the switch (Fig. 2b), suggesting that the global acute and adaptive transcriptional response to carbon source switch does not require Rad53. In agreement with previous findings[16], we observed that a subset of switch-inducible genes was significantly clustered in subtelomeric regions (Fig. 2c). Many of these genes remained de-repressed after long-term ethanol adaptation in *sml1Δ* cells, suggesting a role of their gene products in promoting metabolic adaptation (Fig. 2c, and Supplementary Fig. 2a). We therefore asked if, in *sml1Δrad53Δ* cells, excess histones affected the expression of subtelomeric switch-inducible genes by altering the subtelomeric chromatin context. We found that *sml1Δrad53Δ* cells exhibited aberrant subtelomeric repression of a fraction of switch-inducible genes across several chromosomes (Fig. 2c, Supplementary Fig. 2a, b). The repressed genes included regulators of transporter trafficking (13), metabolic enzymes and regulators (15), and stress resistance factors (7), most of which were ethanol-inducible (Supplementary Fig. 2a).

We next asked if Rad53 regulated the general expression of subtelomeric switch response genes, or specifically their inducibility after glucose starvation. We found that subtelomeric genes were also preferentially repressed in *sml1Δrad53Δ* cells culture in glucose, suggesting that a repressive context was already established before the carbon source switch. Consistent with a general subtelomeric repression, *sml1Δrad53Δ* mutants were also sensitive to rapamycin and chlorpromazine, which both trigger subtelomeric stress responses[16] (Supplementary Fig. 2c). Importantly, deletion of *HHT2* relieved the repression of many subtelomeric genes, both in glucose and after an acute switch to ethanol (Fig. 2d).

We asked how histones increased subtelomere repression. The SIR2/3/4 (SIR, silent information regulator) complex contributes to establish subtelomeric heterochromatin. Deletion of *SIR2*[SIRT1] rescued the repression of subtelomeric (*COS1*, *COS8*) genes in *sml1Δrad53Δ* mutants, both in glucose and after an acute switch to ethanol (Fig. 2e) Notably, we observed that Sir3 mRNA and protein levels were consistently elevated in *sml1Δrad53Δ* cells in an *HHT2*-dependent manner (Supplementary Data 1, Fig. 2f), supporting the idea that elevated SIR expression was involved in establishing histone-induced repressive chromatin. Our data therefore suggest that excess histones in *sml1Δrad53Δ* cells dampen the ethanol switch response in a SIR complex-dependent manner by inhibiting the general expression of subtelomeric genes, and thereby their full inducibility.

Mpk1[ERK5] mediates subtelomere de-repression after rapamycin, by phosphorylating and inactivating Sir3 (Fig. 2g). We therefore asked if glucose starvation acted through the same mechanism, and analyzed Sir3 phosphorylation, which can be visualized as mobility shift in western blot analysis. In contrast to rapamycin, switching cells from glucose to ethanol did not alter Sir3 mobility (Fig. 2g). Hence, the Mpk1–Sir3 pathway is specific to rapamycin and not carbon source switch. We also tested if Rad53 was required for the Mpk1–Sir3 pathway. Sir3 phosphorylation after rapamycin was equally efficient in *sml1Δ* and *sml1Δrad53Δ* cells whereas it was completely blocked in *sml1Δmpk1Δ* controls (Supplementary Fig. 2d). Thus, the Rad53–Spt21 and Mpk1 axes independently modulate subtelomeric expression through the SIR complex.

**Subtelomere expression mediates glucose starvation tolerance.**
We next asked if subtelomeric repression caused glucose dependence. Deletion of *SIR2*, *SIR3*, and *SIR4* or inhibition of Sir2 with nicotinamide partially rescued the hypersensitivity of *sml1Δrad53Δ* cells to glucose restriction (Fig. 3a, Supplementary Fig. 3a, b). In contrast, deletion of *SIR1*, which does not participate in subtelomeric silencing, did not affect glucose dependence (Supplementary Fig. 3c). Rpd3[HDAC1/2/3/8], the catalytic subunit of Rpd3S and Rpd3L HDAC complexes, prevents subtelomeric hyper-silencing in wt cells[17]. We found that *rpd3Δ* mutants with pronounced propagation of subtelomeric silencing showed Sir2-dependent sensitivity to glucose restriction, similar to *sml1Δrad53Δ* mutants (Supplementary Fig. 3d). Thus, our data suggest that DDR- and sirtuin-mediated chromatin regulation facilitate adaptation to glucose restriction by defining subtelomeric heterochromatin propagation.

We next asked how subtelomeric expression could protect from glucose restriction and characterized processes activated under glucose restriction in *sml1Δ* and *sml1Δrad53Δ* cells. *sml1Δrad53Δ* cells did not show abnormal autophagy, mitochondrial protein expression, reduced energy charge or protein synthesis rate (Supplementary Fig. 4a–d). As the respiratory metabolism operated under glucose restriction increases oxidative stress, DDR mutants are sensitive to endogenous oxidative stress[33] and subtelomeres harbor genes involved in redox metabolism (Supplementary Fig. 2a), we measured redox-sensitive metabolites by mass spectrometry. We found that *sml1Δrad53Δ* cells had a lower NADH/NAD+ ratio than *sml1Δ* cells (Fig. 3b), indicative of a more oxidative environment. This was associated with lower levels of the endogenous anti-oxidant *N*-acetylcysteine (NAC) and oxidized glutathione (Fig. 3b), implying less endogenous antioxidant potential. Importantly, *SIR2* deletion partially reverted these metabolite alterations (Fig. 3b). Exogenously applied NAC alleviated the slow growth of *sml1Δrad53Δ* cells under glucose restriction (Fig. 3c), suggesting that reduced anti-oxidant potential in *sml1Δrad53Δ* cells contributed to glucose dependence. The growth benefit by NAC was larger in *sml1Δrad53Δ* than in *sml1Δsir2Δrad53Δ* cells (Fig. 3c), supporting the idea that subtelomere expression provided tolerance to glucose restriction at least in part through enhancing anti-oxidant potential.

To identify the subtelomeric genetic mediators of glucose dependence, we screened all subtelomeric gene deletion mutants contained in the viable haploid synthetic genetic array screening library (Supplementary Fig. 5a, b). This analysis identified two glucose-dependent mutants (*ymr315wΔ*, *yol162wΔ*). However, both genes were not strongly ethanol-inducible, not repressed in *sml1Δrad53Δ* cells, and their expression was not increased by *SIR2* deletion (Supplementary Fig. 5c). We then deleted 12 large subtelomeric regions enriched in repressed genes (see Supplementary Fig. 2a), but none of the deletions resulted in glucose dependence (Supplementary Fig. 5d). As most subtelomeric genes appear in multiple copies distributed over several different subtelomeres (PAU family, COS family, telomeric repeat helicases, alcohol dehydrogenases, and others), it is likely that

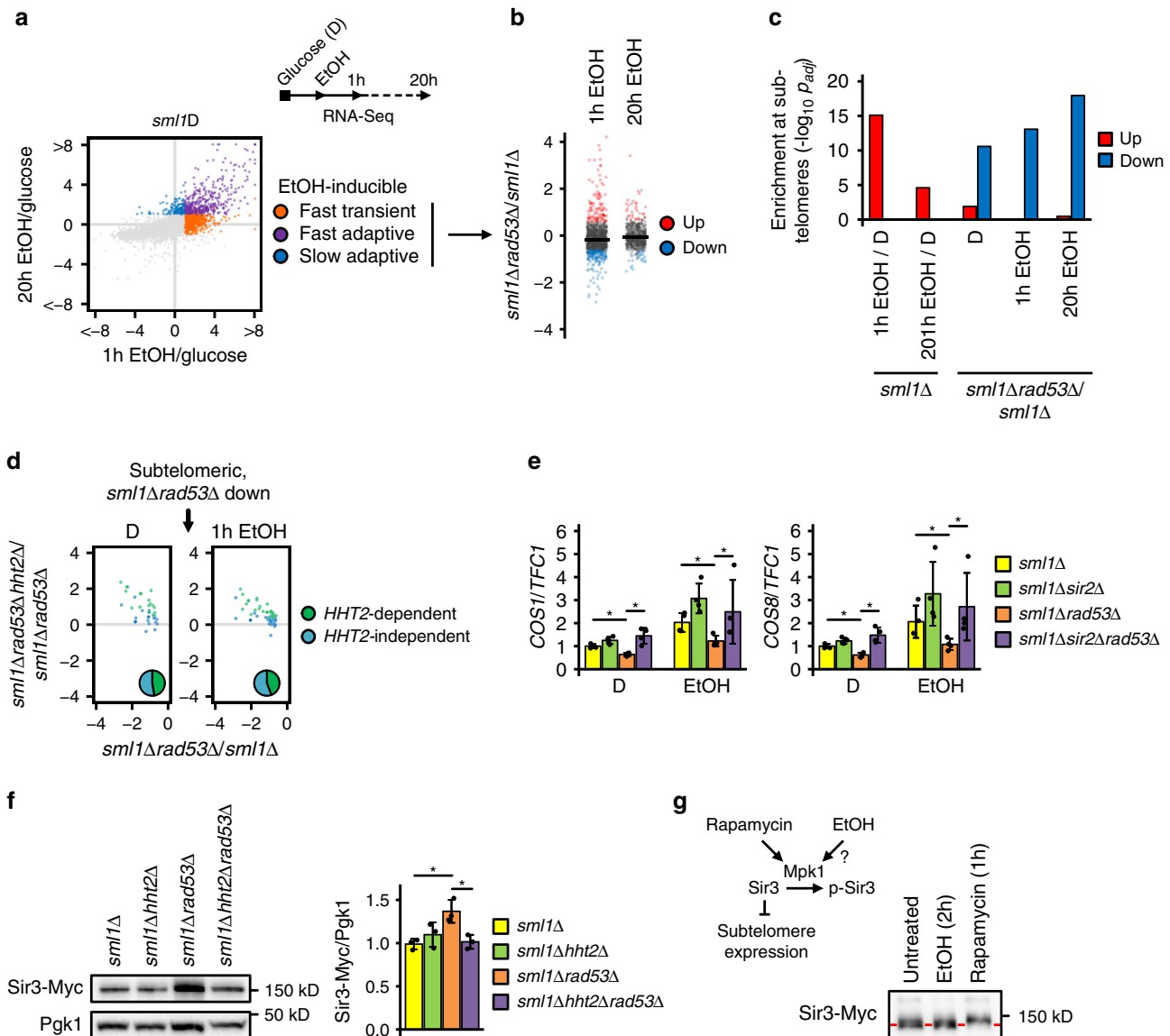

**Fig. 2 Excess histones suppress a subtelomeric starvation response. a** RNA-Seq analysis of carbon source switch in synthetic complete medium. EtOH-inducible genes were categorized as transient or fast/slow adaptive based on their expression at 1 h or 20 h after switching *sml1Δ* cells from glucose to ethanol. **b** Expression fold changes (log2) of switch-induced genes in *sml1Δrad53Δ* vs *sml1Δ* cells at 1 h or 20 h after carbon source switch. Each dot represents one gene. **c** Enrichment analysis by hypergeometric test of genes in telomere proximity that are up- or downregulated by carbon source switch (EtOH/D) in *sml1Δ* cells, or by deletion of *RAD53* (*sml1Δrad53Δ/sml1Δ*). **d** Expression fold changes (log2) of subtelomeric repressed genes in *sml1Δrad53Δhht2Δ* vs. *sml1Δrad53Δ* cells. Each dot represents one gene. **e** RT-qPCR expression analysis of *COS1* and *COS8* genes in the indicated genotypes in glucose and 90 min after ethanol switch. *TFC1* was used as normalization control. ($n = 4$ replicate cultures). Significances were calculated by Kruskal–Wallis rank sum test ($p_{D\_COS1} = 0.0097$, $p_{EtOH\_COS1} = 0.013$, $p_{D\_COS8} = 0.0045$, $p_{EtOH\_COS8} = 0.013$), followed by Mann–Whitney test (two-sided) with Benjamini–Hochberg correction ($p_{D\_COS1\_sml1Δrad53Δ\_vs\_sml1Δ} = 0.029$, $p_{D\_COS1\_sml1Δrad53Δsir2Δ\_vs\_sml1Δrad53Δ} = 0.029$, $p_{EtOH\_COS1\_sml1Δrad53Δ\_vs\_sml1Δ} = 0.029$, $p_{EtOH\_COS1\_sml1Δrad53Δsir2Δ\_vs\_sml1Δrad53Δ} = 0.029$, $p_{D\_COS8\_sml1Δrad53Δ\_vs\_sml1Δ} = 0.029$, $p_{D\_COS8\_sml1Δrad53Δsir2Δ\_vs\_sml1Δrad53Δ} = 0.029$, $p_{EtOH\_COS8\_sml1Δrad53Δ\_vs\_sml1Δ} = 0.029$, $p_{EtOH\_COS8\_sml1Δrad53Δsir2Δ\_vs\_sml1Δrad53Δ} = 0.029$). **f** Cells of the indicated genotypes expressing endogenous 13xMyc-tagged Sir3 were culture in YPD, and the level of Sir3 protein was analyzed by western blot. The bar chart (right panel) provides a quantification of Sir3-Myc levels normalized by loading control Pgk1. Significances were calculated with one-way ANOVA ($p = 0.0104$) with post hoc Tukey HSD test ($p_{sml1Δrad53Δ\_vs\_sml1Δ} = 0.012$, $p_{sml1Δrad53Δhht2Δ\_vs\_sml1Δrad53Δ} = 0.018$). $n = 3$ independent experiments. **g** *sml1Δ* cells expressing endogenous 13xMyc-tagged Sir3 were cultured in normal glucose and subjected to either 2 h glucose-to-ethanol switch or 1 h rapamycin treatment. Sir3 bandshift reflecting phosphorylation was analyzed by western blot. The red dashed line indicates un-phosphorylated Sir3. The scheme summarizes the Mpk1-Sir3 pathway. Bars plots with error bars represent mean values and standard deviation, *D* glucose; *EtOH* ethanol. *$p < 0.05$.

inactivation of a single gene can be compensated for. We hypothesize that the coordinate expression of multiple subtelomeric genes confers glucose starvation resistance (see Supplementary Fig. 2a), but the individual functional contributions remain to be identified.

As subtelomere silencing is coupled with nuclear envelope interaction[34] and *rad53* mutants accumulate perinuclear chromatin[35], we asked if the subtelomeric starvation response was associated with detachment of subtelomeres from the nuclear envelope. We first analyzed the nuclear distribution of silenced

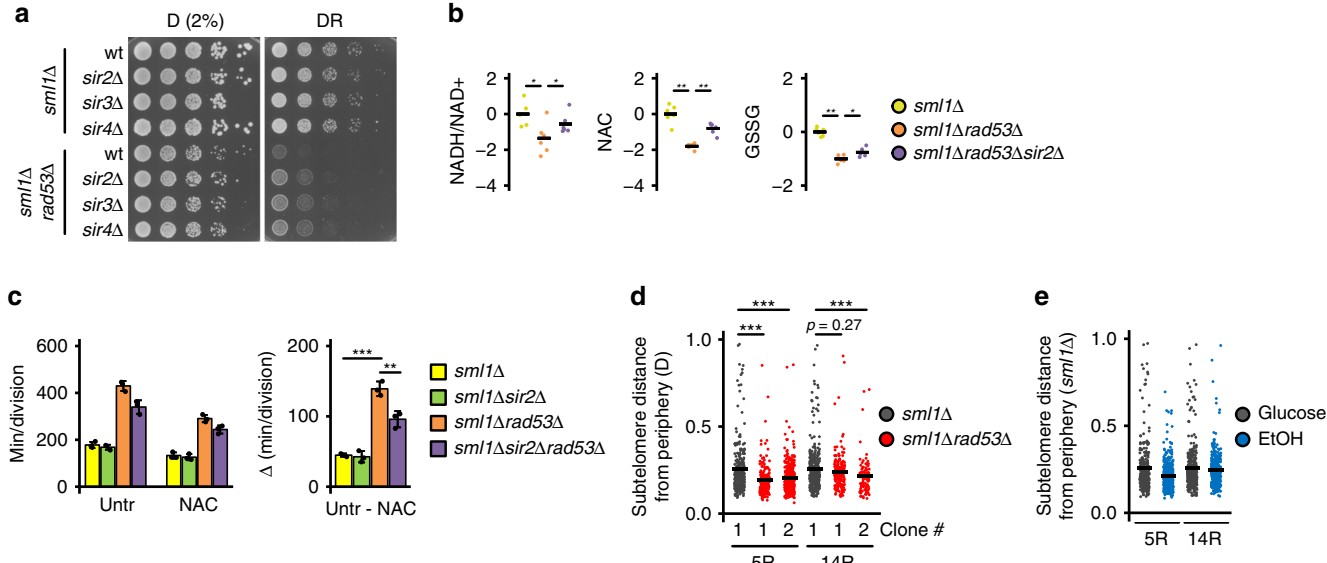

**Fig. 3 Subtelomere expression mediates glucose starvation tolerance. a** $10^7$ cells/mL were serially diluted (1:6), spotted on YP plates with the indicated carbon sources and grown for 2d. **b** Mass spectrometry analysis of redox metabolites in cells of the indicated genotypes cultured in low glucose. Metabolite levels were normalized by the median of all detected metabolites. Significances were calculated by Kruskal–Wallis rank sum test ($p_{NADH/NAD+} = 0.019$, $p_{NAC} = 0.0011$, $p_{GSSG} = 0.0011$), followed by Mann–Whitney test (two-sided) with Benjamini–Hochberg correction ($p_{NADH/NAD+\_sml1\Delta rad53\Delta\_vs\_sml1\Delta} = 0.041$, $p_{NADH/NAD+\_sml1\Delta rad53\Delta sir2\Delta\_vs\_sml1\Delta rad53\Delta} = 0.041$, $p_{NAC\_sml1\Delta rad53\Delta\_vs\_sml1\Delta} = 0.0021$, $p_{NAC\_sml1\Delta rad53\Delta sir2\Delta\_vs\_sml1\Delta rad53\Delta} = 0.0021$, $p_{GSSG\_sml1\Delta rad53\Delta\_vs\_sml1\Delta} = 0.0043$, $p_{GSSG\_sml1\Delta rad53\Delta sir2\Delta\_vs\_sml1\Delta rad53\Delta} = 0.026$). $n = 6$ independent culture replicates. **c** Cells were inoculated in YP + low glucose with or without supply of 1 mM $N$-acetylcysteine, and division speed was determined by cell counting over 2 days. The right panel indicates the reduction of division time by $N$-acetylcysteine. Significances were calculated with one-way ANOVA ($p = 2.4 \times 10^{-6}$) with post hoc Tukey HSD test ($p_{sml1\Delta rad53\Delta\_vs\_sml1\Delta} = 0.0000052$, $p_{sml1\Delta rad53\Delta sir2\Delta\_vs\_sml1\Delta rad53\Delta} = 0.0014$). $n = 3$ independent experiments. **d**, **e** Jitter plots representing the distance of labeled subtelomeres 5 R and 14 R from the nuclear periphery. **d** Cells were cultured in normal glucose. Data from one parental $sml1\Delta$ control and two independent $sml1\Delta rad53\Delta$ clones (#1, #2) are shown. **e** $sml1\Delta$ controls were cultured in the indicated carbon sources. Significances were calculated by Kruskal–Wallis rank sum test ($p_{D\_5R} = 8.5 \times 10^{-15}$, $p_{D\_14R} = 0.00039$), followed by Mann–Whitney test (two-sided) with Benjamini–Hochberg correction ($p_{D\_5R\_sml1\Delta rad53\Delta\#1\_vs\_sml1\Delta} = 3.1 \times 10^{-13}$, $p_{D\_5R\_sml1\Delta rad53\Delta\#2\_vs\_sml1\Delta} = 6.8 \times 10^{-10}$, $p_{D\_14R\_sml1\Delta rad53\Delta\#1\_vs\_sml1\Delta} = 0.27$, $p_{D\_5R\_sml1\Delta rad53\Delta\#2\_vs\_sml1\Delta} = 8.9 \times 10^{-5}$). $n$ cells from X independent experiments ($n$/X): subtelomere 5R D_$sml1\Delta$ = 301/3, D_$sml1\Delta rad53\Delta$#1 = 181/2, D_$sml1\Delta rad53\Delta$#2 = 353/3, EtOH_$sml1\Delta$ = 419/3, subtelomere 14R D_$sml1\Delta$ = 368/2, D_$sml1\Delta rad53\Delta$#1 = 209/2, D_$sml1\Delta rad53\Delta$#2 = 104/1, EtOH_$sml1\Delta$ = 415/2. Bars plots with error bars represent mean values and standard deviation, $D$ glucose; $DR$ glucose restriction (0.01% in solid media, 0.04% in liquid media), $EtOH$ ethanol; $NAC$ $N$-acetylcysteine; $GSSG$ oxidized glutathione. *$p < 0.05$, **$p < 0.01$, ***$p < 0.001$.

subtelomeric regions on the right arms of chromosomes 5 and 14 in one $sml1\Delta$ and two $sml1\Delta rad53\Delta$ clones (Fig. 3d). Consistent with subtelomere silencing, we found that both regions were closer to the nuclear periphery in $sml1\Delta rad53\Delta$ mutants (5 R: significance for 2 clones, 14 R: significance for 1 clone) (Fig. 3d). However, carbon source switch did not release subtelomeres from the nuclear envelope in $sml1\Delta$ cells (Fig. 3e). Hence, repression and peripheral localization of subtelomeres correlate in $sml1\Delta rad53\Delta$ mutants, but release of subtelomeres from the nuclear periphery does not seem to be part of the carbon source switch response.

**Acetylation turnover contributes to glucose dependence.** Although subtelomeric repression accounts in part for histone-mediated glucose dependence, we observed that $sml1\Delta rad53\Delta sir2\Delta$ mutants are not as resistant to glucose restriction as $sml1\Delta rad53\Delta hht2\Delta$ (see Figs. 1b and 3a). Hence, we hypothesized that an additional, potentially Sir-independent mechanism links excess histones to glucose dependence. To identify this mechanism, we screened a yeast H3/H4 histone point mutant library[36] with a $sml1\Delta rad53\Delta$ query strain for functional histone sites involved in glucose restriction tolerance (Supplementary Fig. 6a, left panel). We identified 14 suppressors (Supplementary Fig. 6a, right panel), of which one affects SIR3 binding (H4-T80A)[37] and another likely affects H3 levels through degradation (H3-Y99A)[28].

Interestingly, the substitution of the N-terminal lysine residues 4, 9, 14, and 18 in histone H3 with either glutamine or arginine reduced glucose dependence (Supplementary Fig. 6a, right panel). As glutamine and arginine structurally mimic opposite states of lysine acetylation (acetylated, un-acetylated), this observation suggests that the suppression mechanism is not mediated by the H3 acetylation state per se. Rather, one possibility is that the suppression arises from the inability of HATs to acetylate these residues, resulting in an overall decrease of H3 acetylation rate. Consistently, the H3 acetylation turnover is short (10 min), suggesting that excess H3 would rapidly increase acetylation turnover[38]. We validated the rescue of glucose dependence by substituting the lysine residues 4, 9, 14, 18, and 27 of the endogenous $HHT2$ locus in the W303 background[39] with glutamine ($hht2^{5KQ}$) or arginine ($hht2^{5KR}$) (Fig. 4a and Supplementary Fig. 6b). Deletion of $SIR2$ further alleviated glucose dependence (Fig. 4a and Supplementary Fig. 6b) of the histone acetylation site mutants introduced in the $rad53\Delta$ background; this suggests that the glucose dependence of $rad53\Delta$ mutants is influenced by the subtelomeric chromatin state and by the overall rate of histone acetylation.

Although the contributions of $SIR2$ and H3 acetylation sites to glucose dependence are genetically separable, it is possible that mutating H3 acetylation sites affects SIR-mediated silencing. We therefore analyzed the effect of H3 acetylation site mutations on

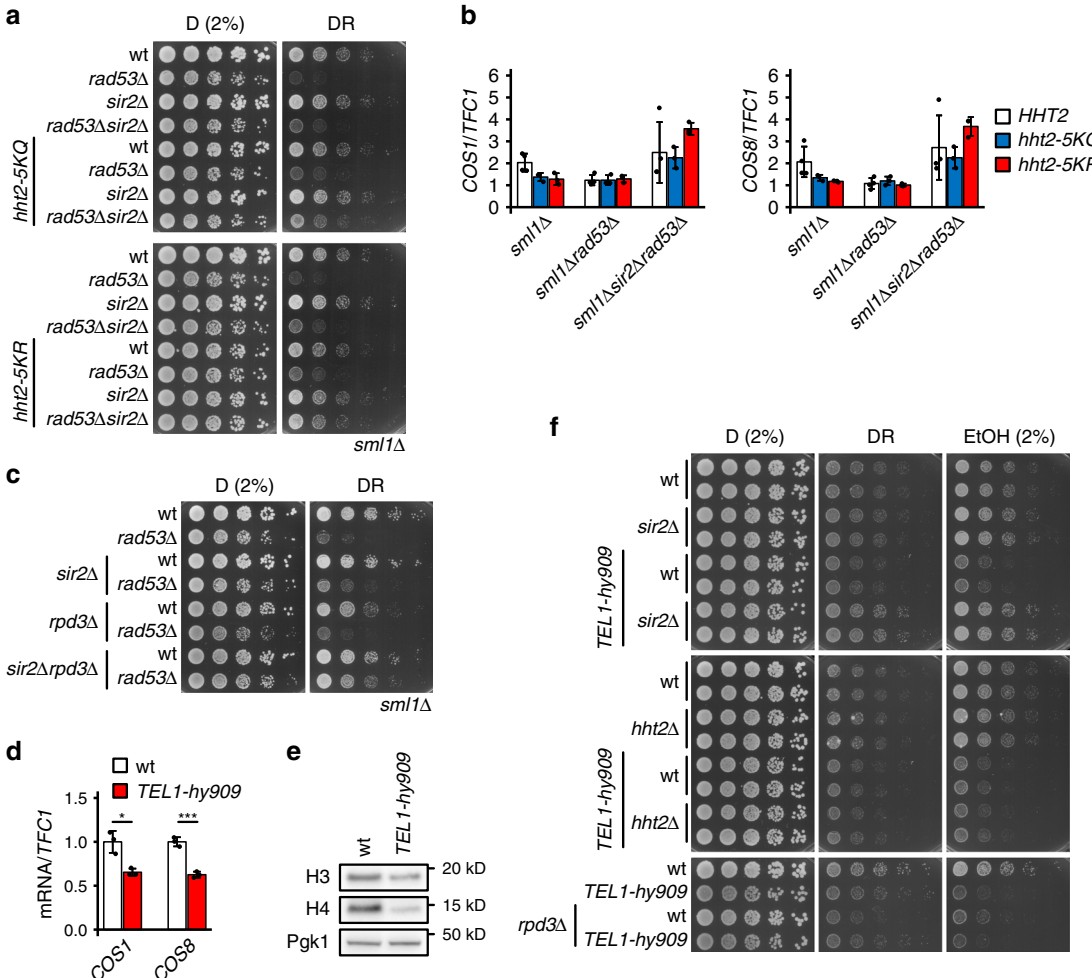

**Fig. 4 Separable effect of histone acetylation turnover and subtelomere silencing on glucose requirement. a** $10^7$ cells/mL were serially diluted (1:6), spotted on YP plates with the indicated carbon sources and grown for 2d. **b** RT-qPCR quantification of the expression of the subtelomeric *COS1* and *COS8* genes in the indicated genotypes 90 min after the switch from glucose to ethanol. Expression levels were normalized to *TFC1*. *n* = 3 (*hht2-5KR* and *hht2-5KQ* mutants) or 4 (*HHT2*) independent culture replicates. **c** $10^7$ cells/mL were serially diluted (1:6), spotted on YP plates with the indicated carbon sources and grown for 2d. **d** RT-qPCR quantification of the expression of subtelomeric genes in wt and *TEL1-hy909* cells grown in YP + 3% ethanol. Expression levels were normalized to *TFC1*. Significance was calculated by Student's *t* test (two-sided, unpaired, $p_{COS1\_TEL1-hy909\_vs\_wt} = 0.030$, $p_{COS8\_TEL1-hy909\_vs\_wt} = 0.00095$). *n* = 3 independent culture replicates. **e** wt and *TEL1-hy909* cells were adapted to YP + 3% ethanol for 20 h and samples were collected during log phase to quantify proteins by western blot analysis. **f** $10^7$ cells/mL were serially diluted (1:6), spotted on YP plates with the indicated carbon sources and grown for 2d. Bar plots with error bars represent mean values and standard deviation, *D* glucose; *DR* glucose restriction (0.01% in solid media), *EtOH* ethanol. *$p < 0.05$, **$p < 0.01$, ***$p < 0.001$.

the expression of the SIR-repressed *COS1* and *COS8* genes. The *hht2^5KR* and *hht2^5KR* mutants did not alter the expression of either SIR-repressed gene in *sml1Δrad53Δ* cells, and only *hht2^5KR* but not *hht2^5KR* enhanced the de-repression of both genes in *sml1Δsir2Δrad53Δ* cells (Fig. 4b). Hence, we do not observe a consistent effect of the *hht2^5KQ* and *hht2^5KR* mutations on SIR-mediated gene repression.

Inhibition of HDACs increases global histone acetylation and consequently reduces the de novo acetylation rate, similar to acetylation-mimicking mutations. We therefore tested whether HDAC deletions alleviated the glucose dependence of *sml1Δrad53Δsir2Δ* cells. Deletion of the catalytic HDAC subunit *RPD3* further rescued glucose dependence, and *sml1Δrad53Δsir2Δrpd3Δ* cells were nearly as resistant to glucose restriction as *sml1Δ* control cells (Fig. 4c). Thus, subtelomere silencing and increased acetylation turnover are two separable mechanisms by which excess histones impose glucose dependence.

**Tel1 affects glucose dependence via subtelomeric chromatin.** In principle, it should be possible to genetically uncouple the telomere effect from the overall acetylation rate in promoting glucose dependence. To this purpose, we investigated an allele specific mutation in *TEL1* (*TEL1-hy909*), which causes constitutive Rad53 activation and telomere over-elongation[40] that could enhance subtelomeric silencing[18]. *TEL1-hy909* mutants exhibited subtelomeric silencing of the SIR-repressed switch response genes (Fig. 4d). Consistent with Rad53 hyper-activation[40], histone levels were low in *TEL1-hy909* mutants (Fig. 4e). *TEL1-hy909* cells were also sensitive to glucose restriction (Fig. 4f). We then addressed whether in *TEL1-hy909* mutants the expression of the subtelomeric switch response genes was limiting, by deleting *SIR2*. We found that *sir2Δ* rescued the *TEL1-hy909* sensitivity to glucose restriction (Fig. 4f). However, ablation of *HHT2* or *RPD3* did not rescue the glucose dependence of *TEL1-hy909* mutants (Fig. 4f). We asked if the subtelomeric repression in *TEL1-hy909*

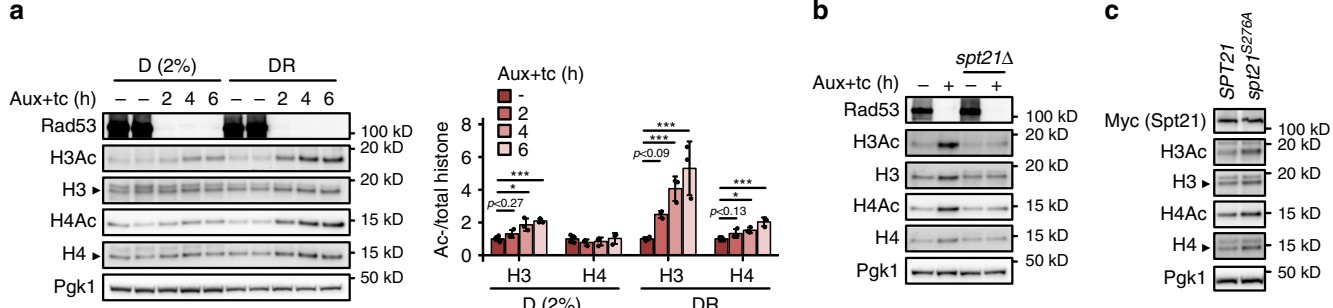

**Fig. 5 Excess histones are hyperacetylated. a** Rad53-AID cells were adapted to normal or low glucose, Rad53 depletion was induced by addition of Auxin and tetracycline, and samples were collected at the indicated times to detect Rad53 and modified histones by western blot analysis ($N = 3$ replicates). The bar chart (right panel) provides a quantification of histone acetylation normalized by total histone levels. Significances were calculated with one-way ANOVA ($p_{D\_AcH3} = 9.4 \times 10^{-5}$, $p_{DR\_AcH3} = 4.0 \times 10^{-5}$, $p_{DR\_AcH4} = 7.4 \times 10^{-5}$) with post hoc Tukey HSD test ($p_{D\_AcH3\_2h\_vs\_UT} = 0.26$, $p_{D\_AcH3\_4h\_vs\_UT} = 0.0010$, $p_{D\_AcH3\_6h\_vs\_UT} = 0.00013$, $p_{DR\_AcH3\_2h\_vs\_UT} = 0.081$, $p_{DR\_AcH3\_4h\_vs\_UT} = 0.00081$, $p_{DR\_AcH3\_6h\_vs\_UT} = 0.000041$, $p_{DR\_AcH4\_2h\_vs\_UT} = 0.12$, $p_{DR\_AcH4\_2h\_vs\_UT} = 0.010$, $p_{DR\_AcH4\_2h\_vs\_UT} = 0.000043$). **b** Rad53-AID cells with the indicated *SPT21* genotype were adapted to low glucose and treated as indicated with Auxin and tetracycline for 4h to deplete Rad53. Samples were collected to detect modified histones by western blot analysis. **c** Spt21-AID and Spt21$^{S276A}$-AID cells were adapted to low glucose and Auxin for 16 h. Auxin was washed away to stabilize Spt21 and samples were collected after 6 h to quantify proteins by western blot analysis. Bars plots with error bars represent mean values and standard deviation, *D* glucose; *DR* glucose restriction (0.02% in liquid media); *Aux* Auxin; *tc* tetracycline. *$p < 0.05$, ***$p < 0.001$.

mutants sensitized to subtelomere-inducing stressors in general. Indeed, *TEL1-hy909* mutants were sensitive to rapamycin in a *SIR2*-dependent manner (Supplementary Fig. 6c), suggesting a general role in subtelomeric gene expression. We conclude that, even with a functional Rad53-Spt21-Histone axis, the subtelomeric silencing of switch response genes can contribute to glucose dependence. Moreover, our observations implicate the Tel1 kinase in modulating glucose dependency through the control of subtelomeric chromatin.

**Histone hyper-acetylation affects central carbon metabolism.** We next asked how excess histones could affect glucose dependence through acetylation turnover. Acetylation requires Ac-CoA as acetyl donor, and the level of Ac-CoA depends on glucose availability[11]. Therefore, we decided to investigate the acetylation state of excess histones. Using an anti-acetyl lysine antibody (T52), we found that global acetylation of H3 and H4 strongly increased within 2 h after Rad53 depletion, in spite of glucose restriction (Fig. 5a). Acetylation increased several-fold more than histone levels, implying that the excess histones were preferentially acetylated. Consistently, preventing histone overproduction by deletion of *SPT21* completely abolished the hyper-acetylation (Fig. 5b), and Spt21$^{S276A}$-driven histone overproduction under glucose restriction was sufficient to induce high acetylation (Fig. 5c). These findings suggest that hyper-acetylation can be a direct consequence of excess histones. We therefore hypothesized that, in *rad53* mutants experiencing glucose restriction, the increased H3/H4 levels, accompanied by their massive acetylation, could further affect the already limiting pool of Ac-CoA and, consequently, the Ac-CoA-dependent anabolic metabolism.

We analyzed the metabolic consequences owing to Rad53 defects. Consistent with carbon metabolite limitation, *sml1Δrad53Δ* cells spontaneously de-repressed glucose starvation genes (Supplementary Fig. 7a, b), showed histone-dependent induction of targets of metabolic (Mig1, Nrg1) and stress response (Msn2/4) TFs (Supplementary Fig. 7c), and shared transcriptome signature similarity with mutants of the glucose-activated Ras signaling pathway (*srv2, ram1Δ*)[41] (Supplementary Fig. 7d).

We analyzed the metabolome of *sml1Δ*, *sml1Δrad53Δ*, *sml1Δhht2Δ*, and *sml1Δrad53Δhht2Δ* cells cultured in a synthetic medium with glucose as sole carbon source. We found that *sml1Δrad53Δ* cells showed global metabolite alterations, which were rescued in *sml1Δrad53Δhht2Δ* cells and therefore dependent on excess histones (Fig. 6a, b). Specifically, *sml1Δrad53Δ* cells showed histone-dependent reduction of central carbon metabolites, including Ac-CoA, glycolysis intermediates and Ac-CoA-derived fatty acids (Fig. 6b–d), and an accumulation of the storage carbohydrate trehalose (Supplementary Fig. 7e). To test if excess histones were sufficient to induce the observed alterations, we acquired the metabolic profile of *sml1Δspt21$^{S276A}$* cells. Indeed, the metabolite alterations in *sml1Δspt21$^{S276A}$* and *sml1Δrad53Δ* cells were overall similar (Supplementary Fig. 8a). In particular, we observed reduced downstream glycolysis intermediates, Ac-CoA and Ac-CoA-derived fatty acids in *sml1Δspt21$^{S276A}$* mutants (Fig. 6c, d, and Supplementary Fig. 8b).

As central carbon metabolites are derived from glucose through glycolysis, we asked if a reduction of glycolysis rate contributed to the low levels of central carbon metabolites. We performed $^{13}C$-glucose flux analysis of *sml1Δ*, *sml1Δrad53Δ*, and *sml1Δrad53Δhht2Δ* cells, cultured in synthetic medium. Although the label incorporation into pyruvate, the end product of glycolysis, was mildly reduced in *sml1Δrad53Δ* mutants compared to the *sml1Δ* control, the labeling was not restored in *sml1Δrad53Δhht2Δ* cells (Supplementary Fig. 8c). Hence, although Rad53 may influence glycolysis, this effect was independent from histone levels. In yeast, pyruvate is directly decarboxylated in the cytoplasm to yield acetate for Ac-CoA synthesis, and increased cytoplasmic Ac-CoA consumption may reduce its flux into the TCA cycle. We therefore measured the carbon label incorporation into TCA intermediates and found a mild reduction in the labeling efficiency of Citrate and α-Ketoglutarate, which was restored by *HHT2* deletion (Supplementary Fig. 8c). These data show that excess acetylated histones in *sml1Δrad53Δ* cells correlate with reduced glucose flux into the TCA cycle.

We next tested the combined effect of glucose restriction (ethanol, low glucose) and *RAD53* deletion (*sml1Δrad53Δ* vs *sml1Δ*). As expected, glucose restriction in *sml1Δ* cells reduced glycolysis intermediates and fatty acids, and caused an

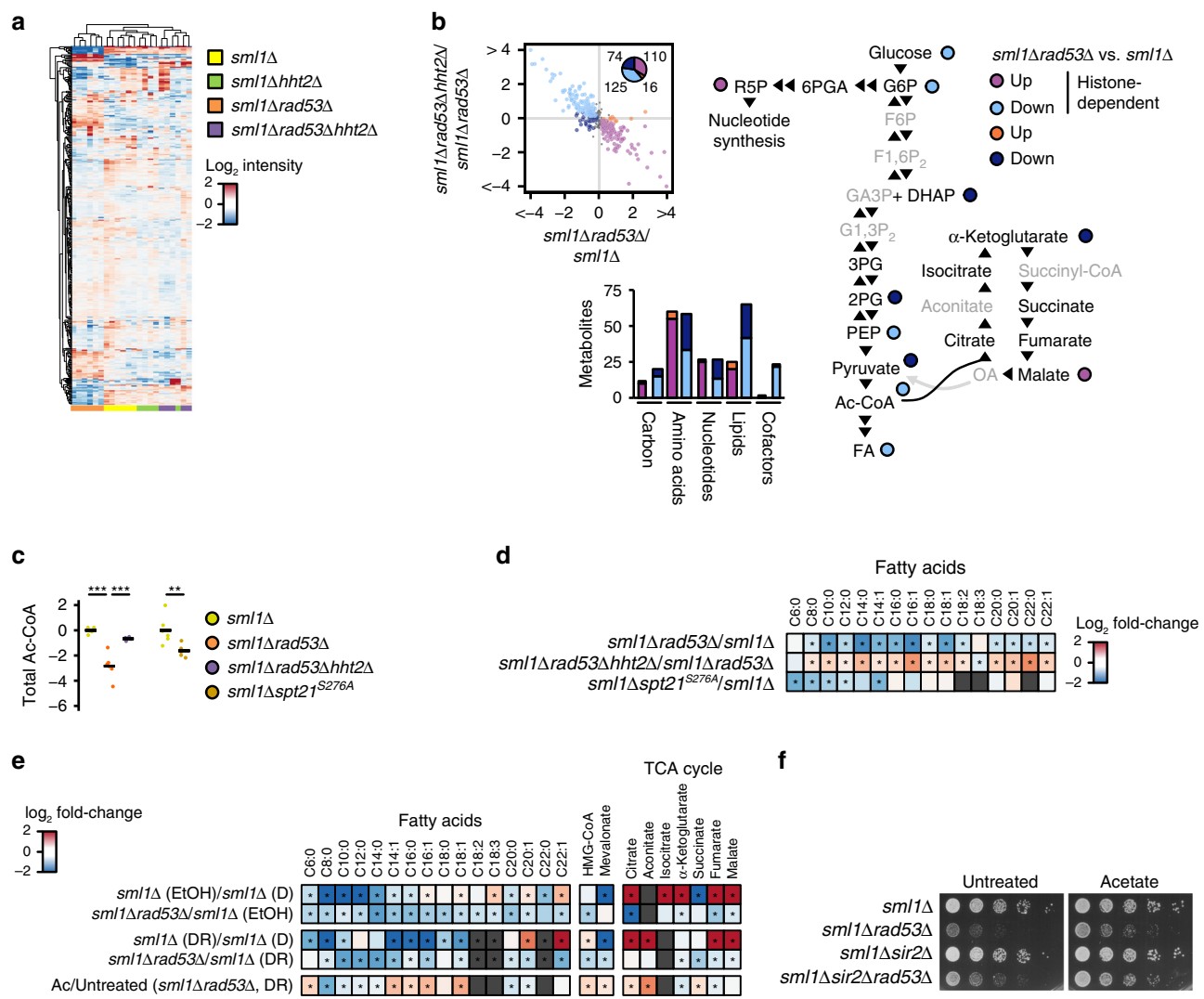

**Fig. 6 Histone hyper-acetylation affects central carbon metabolism. a** Global metabolome analysis of cells of the indicated genotypes cultured in synthetic complete medium with normal glucose. Replicates were clustered by centered log2 metabolite intensities (*n* = 5–6 independent replicate cultures). **b** Classification of altered metabolites in *sml1Δrad53Δ* vs *sml1Δ* cells by *HHT2* dependence and super-pathway. Alterations in glycolysis and TCA cycle are depicted. Significances were calculated by SAM (FDR < 0.1). **c** Jitter plots of total Ac-CoA levels from metabolomics data. The indicated genotypes are normalized by the average of the internal *sml1Δ* control strain. The mean values are represented by black bars. Significances were calculated by Kruskal–Wallis rank sum test ($p_{sml1Δrad53Δhht2Δ\_sml1Δrad53Δ\_sml1Δ}$ = 0.0041), followed by Mann–Whitney test (two-sided) with Benjamini–Hochberg correction ($p_{sml1Δrad53Δ\_vs\_sml1Δ}$ = 0.0043, $p_{sml1Δrad53Δhht2Δ\_vs\_sml1Δrad53Δ}$ = 0.0043), or by Mann–Whitney test ($p_{sml1Δspt21S276A\_vs\_sml1Δ}$ = 0.0022). *n* = 5 (*sml1Δrad53Δhht2Δ*) or 6 (other genotypes) independent culture replicates. **d** Heat map representing fatty-acid alterations from metabolomics data in cells with the indicated experimental vs control genotypes. Asterisks indicate significant alterations by SAM analysis (FDR < 0.1, *q* values are listed in Supplementary Table 4). **e** Heat map representing fatty-acid sterol synthesis and TCA cycle intermediate alterations from metabolomics data in cells with the indicated experimental vs control genotypes or conditions. Asterisks indicate significant alterations by SAM analysis (FDR < 0.1, *q* values are listed in Supplementary Table 5). **f** 10^7 cells/mL were serially diluted (1:6), spotted on YP plates with 0.01% glucose, with or without acetate, and grown for 2d. *D* glucose, *DR* glucose restriction (0.04% in liquid media), *EtOH* ethanol; *Aux* Auxin; *tc* tetracycline; *Ac-CoA* acetyl-coenzyme A; *G6P* glucose 6-phosphate; *F6P* fructose 6-phosphate; *F1,6P₂* fructose 1,6-bisphosphate; *GA3P* glyceraldehyde 3-phosphate; *DHAP* dihydroxyacetone phosphate; *G1,3P₂* 1,3-bisphosphoglycerate; *3PG* 3-phosphoglycerate; *2PG* 2-phosphoglycerate; *PEP* phosphoenolpyruvate; *FA* fatty acids, *6PGA* 6-phosphogluconate; *R5P* ribose 5-phosphate; *HMG-CoA* hydroxymethylglutaryl-Coenzyme A, *TCA* tricarboxylic acid. \*\**p* < 0.01, \*\*\**p* < 0.001.

accumulation of TCA intermediates for mitochondrial respiration (Fig. 6e, Supplementary Fig. 9a, b), and Ac-CoA levels were below the detection limit. Deletion of *RAD53* further enhanced the depletion of Ac-CoA-derived fatty acids, reduced the level Ac-CoA-derived sterol precursors, and reduced the accumulation of a subset of TCA metabolites (Fig. 6e).

Ac-CoA for acetylation and anabolic reactions can be directly and rapidly derived from imported acetate[11]. Hence, we hypothesized that providing acetate as an Ac-CoA source to *sml1Δrad53Δ* cells might alleviate their glucose dependence.

Indeed, acetate supply increased the level of Ac-CoA-derived fatty acids and sterol precursors and ameliorated the sensitivity of *sml1Δrad53Δ* cells to glucose restriction (Fig. 6e, f). This rescue did not correlate with a restoration of the affected TCA cycle intermediates (Fig. 6e), suggesting that replenishment of TCA cycle intermediates was not the cause of enhanced proliferation. Moreover, it further improved the growth of *sml1Δrad53Δsir2Δ* mutants under glucose restriction (Fig. 6f). Hence, our data suggest that limitation of Ac-CoA and some of its derived metabolites contributed to the glucose dependence of *rad53*

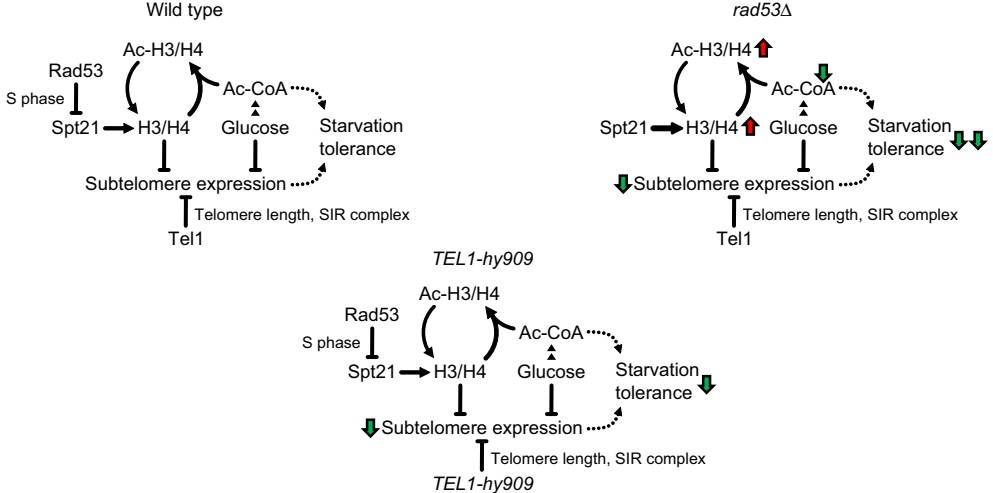

**Fig. 7 Model.** Rad53 regulates histone expression through inhibition of Spt21[NPAT] by phosphorylation. Absence of this regulation causes accumulation of histones. Hyperacetylated excess histones reduce growth-limiting Ac-CoA-derived carbon metabolites. Excess histones also silence starvation-responsive subtelomeric genes. Hyper-acetylation and subtelomeric gene silencing reduce the glucose starvation tolerance of *rad53* mutants. Hyperactive Tel1-hy909 increases telomere length and subtelomere silencing and thereby reduces glucose starvation tolerance in a histone-independent manner. Red and green arrows indicate the directions of regulation.

mutants. In summary, these findings imply that in *sml1Δrad53Δ* cells, excess histones impose glucose dependence by limiting Ac-CoA and Ac-CoA-derived central carbon metabolites.

## Discussion

We show that the DDR kinase Rad53 acts on the conserved histone regulator Spt21[NPAT] to limit histone production. Loss of this regulation results in histone accumulation and acetylation, and accounts for the specific glucose dependence of *rad53* mutants. We further show that excess histones interfere with the tolerance to glucose restriction by two separable mechanisms, the SIR-dependent silencing of starvation-induced subtelomeric domains, and the depletion of central carbon metabolites (Fig. 7).

Previous studies suggested that direct binding of excess histones to Rad53 initiates their degradation[28] and that, following DNA breaks, DDR signaling suppresses histone accumulation[42]. Here we provide evidence of a non-canonical function of the Rad53 kinase in limiting histone levels in the absence of DNA damage insults, and of a baseline activity of the Rad53–Spt21 axis in limiting histone expression during an unperturbed S phase. The model is consistent with studies describing that unperturbed DNA replication is sufficient for basal DDR activation[19,32,43]. This Rad53-mediated regulatory loop is likely integrated in the signal transduction cascade involving the two upstream DDR kinases, Mec1 and Tel1, although further work will be needed to address their specific roles in controlling glucose tolerance. Although the Spt21-histone axis is operated during a physiological S phase, it is possible that its activity is enhanced after DNA damage to prevent histone synthesis. Notably, we consistently observe stronger protein level imbalances of histone H4 than H3 in Rad53–Spt21 axis mutants. However, additional work is required to understand the mechanistic basis and the potential physiological relevance of this finding.

The activities of Spt21 and its human homolog, NPAT, oscillate and peak in S phase when the need for histone synthesis is high[29,44]. Intriguingly, the Rad53 target site on Spt21 (S276) is conserved in NPAT (S981). Although the Rad53 consensus motif changed during evolution, NPAT is regulated by S phase CDK activity[31], which is one of the most important DDR targets. During evolution, NPAT regulation may therefore have shifted from direct to indirect regulation, and complemented with additional regulation layers such as mRNA stability[45].

In budding yeast, subtelomeres constitute the largest known regions in which nuclear envelope attachment is coupled to a constitutive silent chromatin state by the SIR complex[34]. This silent chromatin harbors stress response genes that become active during carbon source limitation[16]. We show that histone control by the Rad53 kinase counteracts SIR-dependent gene silencing to support the response to metabolic stress. As Sir3 levels determine subtelomeric silencing[46], excess histones may mediate subtelomeric silencing in part by elevating Sir3 expression. By using the specific *TEL1-hy909* allele that maintains Rad53 active and histone levels limiting, but causes aberrant subtelomere silencing, we have uncoupled DDR-mediated subtelomeric repression from histone control. The mechanisms by which Rad53 and Tel1 influence subtelomere silencing seem therefore different: while in *rad53* mutants the muffling of subtelomeric switch response genes is caused by exceeding histones, subtelomeric silencing in *TEL1-hy909* mutants likely results from telomere length expansion[40].

The nuclear envelope is a hub for silenced subtelomeric chromatin[47] and our observations imply that chromatin attachment to the nuclear periphery may influence tolerance to glucose starvation. Accordingly, glucose dependence of both *rad53Δ* and *TEL1-hy909* mutants is alleviated by ablating the SIR complex, which is a mediator of chromatin silencing and attachment. These data point out a potential physiological relevance of the nuclear envelope-chromatin association in influencing glucose tolerance. Moreover, in light of recent observations implicating the DDR in controlling chromatin-nuclear envelope attachment, altogether our findings pinpoint a key role for the nuclear envelope in influencing not only replication stress[35] and mechanotransduction[48], but also cell metabolism.

Our observations underline the existence of an integrated network mediated by different DDR kinases that controls histone levels and subtelomeric silencing and couples the epigenetic state of the cell with metabolic changes. In normal cells, subtelomeres may thus sense histones as readout for protein synthesis capacity, which is directly influenced by glucose availability. Our data contribute to unmask a key role for histones in modulating glucose tolerance. In mammalian neurons, histone availability

directly determines activity-dependent transcription, nucleosome re-depositioning, synaptic connectivity, and behavior[49]. Thus, correct histone dosage may similarly dictate metabolic and signaling responses in mammals, but the genomic response domain may depend on the cell type and stimulus.

Our data suggest that increased Ac-CoA consumption directly contributes to metabolic imbalancements in *rad53* mutants. Glucose-derived Ac-CoA is thought to be limiting during glucose restriction[50]. Ac-CoA levels rise in G1 phase until a threshold is exceeded, which allows acetylation of HATs and histones at genes for S phase entry[11]. Surprisingly, we found that cells with low-glucose supply are capable of highly acetylating excess histones, resulting in total acetylation levels, which even exceed those found in high-glucose medium. Our observations suggest that acetylation of excess histones directly consumes Ac-CoA, limits its availability for biosynthetic reactions and induces glucose dependence. In the light of a strict temporal compartmentalization of metabolic cycles[51], control of histone levels may therefore prevent an unscheduled starvation response and metabolic reprogramming during S phase. The replication-coupled histone control thus represents a way to guarantee Ac-CoA availability for replication-associated processes, which becomes essential under conditions of glucose restriction.

We have shown that histone hyper-acetylation and subtelomere silencing are genetically separable causes of glucose dependence, using the combination of H3 acetylation mutants and *SIR2* deletion. However, given the various regulatory roles of H3 and H4 acetylation on subtelomere expression, we assume that histone hyper-acetylation likely influences subtelomere expression in some loci and thus have an additional indirect impact on glucose dependence.

Cancers have an elevated glucose uptake and requirement in comparison with normal tissues, to fuel the biosynthetic pathways driving rapid proliferation[1]. Targeting of glucose metabolism[2] and nutrient restriction[3] may complement classical cancer treatments in the future. In several types of cancer, histone genes are amplified as a consequence of copy number increases of the HIST1 and HIST2 loci[52,53]. Moreover, various cancers exhibit upregulation of specific histone genes[54,55]. Notably, NPAT has an important role in driving replication-dependent histone expression in breast cancers with cyclin E2 overexpression[56], and germline NPAT mutations are a candidate risk factor for Hodgkin lymphoma, highlighting a role for excessive histone expression in tumorigenesis[57]. Based on our findings, we suggest that high histone levels in cancers may sensitize to the targeting of glucose metabolism or nutrient restriction, and may serve as biomarker for the success of interventions targeting glucose metabolism.

## Methods

**Media and drug sensitivity assay**. All yeast strains are listed in Supplementary Table 2. Unless otherwise stated, yeast strains were grown in yeast extract/peptone with 2% glucose at 28 °C. G1 synchronization was done with 3 mg/mL α-factor. For drug sensitivity assay, cells were grown to stationary phase, serial 1:6 dilutions were made, and one drop of each dilution was pin-spotted onto agar plates containing the indicated drugs and carbon sources. Plates were incubated for 2 days at 28 °C. For all liquid growth assays, division time was assessed during log phase growth ($<1.5 \times 10^7$ cells/mL in 2% glucose, $<0.2 \times 10^7$ cells/mL in 0.02% glucose). For glucose restriction experiments on solid media, 0.01% glucose were used on plates in general, and 0.04% were used on plates with the *spt21^S276A* mutant because its lethality on 0.01% glucose does not allow the assessment of genetic interaction with *rad53Δ*. For liquid experiments, 0.02% were used for most experiments to ensure that cells were in logarithmic growth phase during the entire experiment. In all, 0.04% were used for quantitative long-term division time experiments and metabolomics experiments to increase the range of the logarithmic growth phase and obtain sufficient material for analysis.

**Drug treatments**. Drugs were used at the following concentrations. Auxin (Merck Cat# I5148): 1 mM. Tetracycline (nzytech Cat# MB02202): 0.6 mM in 2% glucose, 0.3 mM in 0.02% glucose. Nicotinamide (Merck Cat# N0636): 4 mM. Hydroxyurea (Merck Cat# H8627): as indicated. Rapamycin (Merck Cat# R0395): as indicated. Chlorpromazine (Merck Cat# C8138): as indicated.

**Western blot analysis**. For western blot analysis, cells were fixed with 20% trichloroacetic acid (TCA) and disrupted by bead beating. Lysate and precipitate/debris were mixed with 600 μL 10% TCA and pelleted. The pellet was resuspended in 1× Laemmli buffer with 5% β-mercaptoethanol and 160 mM Tris-HCl (pH 7.4), boiled for 10 min and sonicated briefly. The extract was directly subjected to sodium dodecyl sulfate (SDS) gel analysis without clearance. The following antibodies were used: mouse monoclonal anti-Rad53 (clone EL7, in house[58], 1:5), mouse monoclonal anti c-MYC (clone 9E10, Santa Cruz Biotechnology Cat# sc-40, RRID:AB_627268, 1:2000), rabbit anti-histone H3 (EpiCypher, Cat# 13-0001, 1:5000), rabbit anti-histone H4 (Abcam Cat# 7311, 1:4000), rabbit anti-histone H3 (acetyl K27) (Abcam, Cat# ab4729, 1:2000), mouse monoclonal anti-acetyl-lysine (T52, in-house[59], 1:10), mouse monoclonal anti-Pgk1 (novex, Cat# 459250, 1:10,000), mouse monoclonal anti-Porin (abcam, Cat# 110326, 1:1000), rabbit anti-GFP (Amsbio, Cat# TP401, 1:5000), goat anti-mouse IgG (H + L)-horseradish peroxidase (HRP) Conjugate (Bio-Rad, Cat# 1706516, 1:20,000), goat anti-rabbit IgG (H + L)-HRP Conjugate (Bio-Rad, Cat# 1706515, 1:20,000). Detection was performed by electrochemiluminescence (GE Healthcare). Uncropped western blots are shown in Supplementary Figs. 10 and 11.

**Quantitative PCR analysis**. Total mRNAs were extracted from $2 \times 10^7$ cells using the RNeasy mini kit (QIAGEN). Reverse transcription was performed using the SuperScript VILO cDNA synthesis kit (Invitrogen) with 1 mg of total RNA. In all, 1/80 of the cDNA reaction was used for quantitative PCR. Quantitative PCR was prepared using QuantiFast SYBR green PCR kit (QIAGEN) and run on the LightCycler 96 (Roche Life Science) according to the manufacturer's instructions. Relative cDNA quantification was performed. *TFC1* was chosen as normalization target for its robust correlation with the median of all mRNAs across genotypes and conditions in the RNA-Seq experiments. Primers used for quantitative PCR are listed in Supplementary Table 3.

**Identification of phosphorylation sites by mass spectrometry**. Cells were grown in CSM (–)Arg (–)Lys minimal media supplemented with light and heavy-labeled amino acids (light version complemented with normal arginine and lysine; heavy version complemented with L-lysine 13C6, 15N2. HCl and L-arginine 13C6, 15N4.HCl). Light and heavy-labeled cultures were combined, harvested by centrifugation in TE buffer pH 8.0 containing protease inhibitors and stored frozen at −80 °C until cell lysis. Approximately 0.3 g of yeast cell pellet (in three separate 2 mL screwcap tubes) was lysed by bead beating at 4 °C in 3 mL of lysis buffer (1 mL per tube) containing 50 mM Tris-HCl, pH 8.0, 0.2% Tergitol, 150 mM NaCl, 5 mM ethylenediaminetetraacetic acid (EDTA), complete EDTA-free protease inhibitor cocktail (Roche), 5 mM sodium fluoride and 10 mM β-glycerophosphate. Lysates of light and heavy conditions were mixed together (~6 mg of protein from each condition). The mixed lysate was denatured in 1% SDS, reduced with DTT, alkylated with iodoacetamide and then precipitated with three volumes of a solution containing 50% acetone and 50% ethanol. Proteins were solubilized in a solution of 2 M urea, 50 mM Tris-HCl, pH 8.0, and 150 mM NaCl, and then TPCK-treated trypsin was added. Digestion was performed overnight at 37 °C, and then trifluoroacetic acid and formic acid were added to a final concentration of 0.2%. Peptides were desalted with Sep-Pak C18 column (Waters). C18 column was conditioned with five column volumes of 80% acetonitrile and 0.1% acetic acid and washed with five column volumes of 0.1% trifluoroacetic acid. After samples were loaded, column was washed with five column volumes of 0.1% acetic acid followed by elution with four column volumes of 80% acetonitrile and 0.1% acetic acid. Elution was dried in a SpeedVac evaporator and resuspended in 1% acetic acid. Desalted peptides were resuspended in 1% acetic acid and loaded in a tip column containing ~22 μl of immobilized metal affinity chromatography (IMAC) resin[32]. After loading, the IMAC resin was washed with one column volume of 25% acetonitrile, 100 mM NaCl, and 0.1% acetic acid solution followed by two column volumes of 1% acetic acid, one column volume of deionized water and finally, eluted with three column volumes of 12% ammonia and 10% acetonitrile solution. The elutions were dried in a SpeedVac, reconstituted in 80% acetonitrile and 1% formic acid and fractionated by hydrophilic interaction liquid chromatography (HILIC) with TSK gel Amide-80 column (2 mm × 150 mm, 5 μm; Tosoh Bioscience). In all, 90 sec fractions were collected between 10 and 25 min of the gradient. Three solvents were used for the gradient: buffer A (90% acetonitrile); buffer B (80% acetonitrile and 0.005% trifluoroacetic acid); and buffer C (0.025% trifluoroacetic acid). The gradient used consists of a 100% buffer A at time = 0 min; 88% of buffer B and 12% of buffer C at time = 5 min; 60% of buffer B and 40% of buffer C at time = 30 min; and 5% of buffer B and 95 % of buffer C from time = 35–45 min in a flow of 150 μl/min. HILIC fractions were dried in a SpeedVac, reconstituted in 0.1% trifluoroacetic acid and subjected to LC-MS/MS analysis using a 20-cm-long 125-μm inner diameter column packed in-house with 3 μm C18 reversed-phase particles (Magic C18 AQ beads, Bruker). Separated phosphopeptides were electrosprayed into a QExactive Orbitrap mass spectrometer (Thermo Fisher Scientific). Xcalibur 3.1.66.10 (Thermo Fisher Scientific) was used

for the data acquisition and the Q Exactive was operated in data-dependent mode. Survey scans were acquired in the Orbitrap mass analyzer over the range of 380–1800 m/z with a mass resolution of 70,000 (at m/z 200). MS/MS spectra was performed selecting up to the 10 most abundant ions with a charge state using of 2, 3, or 4 within an isolation window of 2.0 m/z. Selected ions were fragmented by higher-energy collisional dissociation with normalized collision energies of 27 and the tandem mass spectra was acquired in the Orbitrap mass analyzer with a mass resolution of 17,500 (at m/z 200). Repeated sequencing of peptides was kept to a minimum by dynamic exclusion of the sequenced peptides for 30 seconds. For MS/MS, AGC target was set to 1e5 and max injection time was set to 120 ms. All spectra were also searched using two Sequest-based engine SORCERER 5.1.1 (Sage N Research, Inc.). Precursor match tolerance for both Sequest searches was set to 10 ppm. Differential modifications were 8.0142 daltons for lysine, 10.00827 daltons for arginine, 79.966331 daltons for phosphorylation of serine, threonine, and tyrosine, phosphorylation dehydration, and a static mass modification of 57.021465 daltons for alkylated cysteine residues We only considered PSMs whose peptide prophet score exceeded 0.7. Two replicate label swap experiments were performed and representative results were depicted. Only phosphopeptides identified in both experiments that represent Rad53-dependent phosphorylation events (as defined in Bastos de Oliveira et al., 2015) were included in the plot (Supplementary Table 1). All spectra that were identified only one time were manually inspected.

**Alignment of protein sequences**. Budding yeast Spt21 and human NPAT were aligned with blastp using default settings. The alignment was visualized with Jalview 2.10.5[60] using Clustal coloring.

**RNA-Seq analysis**. Libraries for RNA sequencing were prepared following the manufacturer protocols for transcriptome sequencing with the ion proton sequencer (Thermo Fisher Scientific/Life Technologies). In brief, 1 µg of total RNA was poly-A selected using the dynabeads mRNA direct micro purification kit (Thermo Fisher Scientific, Cat# 61021) according to manufacturer's protocol. In all, 50 ng of poly-A RNA were used to prepare strand-specific barcoded RNA libraries with the Ion Total RNA-Seq kit v2.0 (Thermo Fisher Scientific, Cat# 4475936). Poly-A RNA was fragmented with RNAse III, and purified with nucleic acid binding beads. RNA fragments were hybridized and ligated with ion adapter and subsequently reverse transcribed into cDNA. The cDNA was amplified with the Ion Torrent barcode primer and purified with nucleic acid binding beads. Final libraries were quantified on the Qubit fluorometer with HS DNA (Thermo Fisher Scientific/Life Technologies) and checked for size on an Agilent Bioanalyzer with HS DNA kit (Agilent, Santa Clara, CA). Ten barcoded libraries were pooled together on an equimolar basis at a final concentration of 11 pM and clonally amplified using the Ion Proton Hi-Q Template Kit (Thermo Fisher Scientific/Life Technologies) with IonOneTouch 2 instrument (Thermo Fisher Scientific/Life Technologies). After emulsion PCR, DNA-positive ISPs were recovered and enriched according to standard protocols with the IonOneTouch ES Instrument (Thermo Fisher Scientific/Life Technologies). A sequencing primer was annealed to DNA-positive ISPs and the sequencing polymerase bound, prior to loading of ISPs into Ion P1-sequencing chips. Sequencing of the samples was conducted according to the Ion Proton Hi-Q Sequencing Kit Protocol on Ion Proton instrument.

Raw reads were converted to reads per ORF on a Galaxy Server (https://usegalaxy.org/). In brief, BAM files were converted to FASTQ using BEDTools, and reads were annealed to the SacCer_Apr2011/sacCer3 transcriptome using Sailfish (k-mer size: 21, bias correction). The R package DESeq2 (version 1.16.1) was used to identify differentially expressed ORFs by pairwise comparison (fitType = local, pAdjustMethod = BH). ORFs with <50 reads in both conditions were excluded before significance analysis. A BH-adjusted p value (FDR) of 0.1 was used to identify differentially expressed ORFs. Fold-change thresholds were applied for grouping of significantly altered ORFs into signatures (fourfold for global definition of strongly ethanol-inducible genes, 1.5-fold for differential expression and ethanol-induced genes for mapping in subtelomeres, no threshold for rescue of differential expression). ORF information was obtained from the R package org.Sc.sgd.db (version 3.4.1). For enrichment of ORFs in telomere proximity, a distance of 40 kb from the center of each ORF to the chromosome end was used. TF target enrichment was based on a curated data set and TFs with <20 targets were excluded[61]. Significance in proximity and TF target enrichment analyses was determined by hypergeometric test (BH-corrected p value < 0.05). For literature data set correlation analysis, the SPELL tool (2.0.3)[62] was used to identify relevant data sets, which were obtained from SGD (https://spell.yeastgenome.org/search/dataset_listing). STRING (11.0) and Cytoscape (3.5.1) were used for network clustering and visualization.

**Imaging of subtelomere localization**. Strains containing fluorescently labeled subtelomeres with 128 Lac operators inserted at the sites of interest and constitutive expression of LacI repressor fused to eGFP[63] were grown in synthetic complete medium with 2% glucose or 3% ethanol to 1 OD for 20 h at 30 °C. Cells were embedded in agarose patches composed of the same culture medium and sealed using VaLaP (1/3 Vaseline, 1/3 Lanoline, and 1/3 Paraffin). Live cell imaging was performed with a wide field microscopy system featuring a Nikon Ti-E body equipped with the Perfect Focus System and a ×60 oil immersion objective with a

numerical aperture of 1.4 (Nikon, Plan APO), an Andor Neo sCMOS camera (field of view of 276 × 233 lm at a pixel size of 108 nm, Andor Driver Pack 3) and LEDs (SpectraX) controlled by the NIS-Elements software 4.60. 3D z-stacks consisting of 25 frames with z-steps of 300 nm were acquired using an eGFP filter with an exposure time of 200 ms. To precisely measure the distance between the spot and the nuclear envelope, images were first deconvolved using iterative Deconvolve 3D plugin and pointing of nuclear envelope edge, locus and opposite side of the nuclear envelope was performed using point picker tool from Image J on a single focal plane. Nucleus diameter was estimated and relative distance between the spot and the edge (at the envelope=0%) was calculated.

**Histone point mutant screen**. Histone H3 and H4 point mutants on Lys, Ser, Thr, and Tyr resides were selected from a library[36], and crossed with a sml1Δ rad53Δ HHT1-HHF1-LEU2 query strain. Diploids were selected on synthetic complete medium without uracil and leucine, and with 200 µg/mL G418 (ThermoFisher Cat# 11811031) for 2 days. Diploids were sporulated on VB dishes (100 mM NaAc, 25 mM KCl, 20 mM NaCl, 3 mM MgSO$_4$, 1.5% agar) for 8 days. Spores were streaked on synthetic complete medium without uracil, leucine, histidine, arginine, lysine, and ammonium sulfate, and with 0.1% monosodium glutamate, 200 µg/mL G418, 25 µg/L canavanine (Merck Cat# 861839-1G), and 25 µg/L thialysine (Merck Cat# A2636-1G). Three clones were selected and glucose dependence was assessed by spot assay.

**Metabolite analysis**. Yeast cells were grown to logarithmic phase ($1 \times 10^7$ cells/mL) in synthetic complete medium with either 2% glucose or 3% ethanol as carbon source. In all, $5 \times 10^8$ cells were harvested by centrifugation, washed in water, and snap-frozen in liquid nitrogen. Metabolite extraction and ultrahigh performance liquid chromatography-tandem mass spectroscopy analysis were performed by Metabolon (Durham, North Carolina) as previously described[23]. Data analysis was performed with the software RStudio (1.0.153). Missing metabolite raw intensity values were imputed with the lowest detectable value, and intensities were median-normalized. Fold changes and significant alterations were calculated with the SAM method[64] implemented in the R package samr. Clustering and heat map representation were done with the R package pheatmap. An FDR threshold of 0.1 was used to identify altered metabolites.

**Flux analysis**. Yeast cells grown in the log phase with 2% unlabeled glucose were spiked with equal amount of medium having 13 C labeled glucose (all carbons labeled). The metabolites from these cells were extracted and subjected to LC-MS/MS based analysis[65]. In brief, at the indicated time-points, cells were quenched for 5 min in four volumes of 60% methanol (−45 °C), centrifuged at 1000 g (−5 °C) for 3 min and the pellet was washed with 700 µl of 60% methanol (−45 °C) with final centrifugation at 1000 g (−5 °C) for 2 min. For metabolite extraction, the obtained pellet was resuspended in 1 ml 75% ethanol, kept at 80 °C for 3 min, immediately followed by incubation on ice for 5 min and centrifugation at 20,000 g for 10 min. The extracted metabolites were vacuum dried, and stored at −80 °C till further use. For measurements of pyruvate and other TCA metabolites, the extracts were derivatized with O-benzylhydroxylamine (OBHA), with 1-ethyl-3-(3-dimethylaminopropyl) carbodiimide as a catalyst. The OBHA-derivatized products were extracted using ethyl acetate and dried using a vacuum concentrator. The vacuum-dried sample was dissolved in 1 ml of 1:1 water:methanol, and a suitable dilution was injected into the Synergi Fusion-RP column (150 mm). The derivatized products were detected on Agilent's 1290 infinity series UHPLC system coupled to Thermo TSQ Vantage mass spectrometer. The standards were used for developing multiple reaction monitoring methods. The area under each peak was extracted using Thermo Xcalibur software 2.2 SP 1.48 (Qual and Quan browsers).

**Statistics and reproducibility**. The following tests were applied: one-way analysis of variance with post hoc Tukey HSD test for multiple comparisons between groups of normally distributed, parametric data. Student's t test (two-sided, unpaired) for pairwise comparisons between groups of normally distributed, parametric data. Kruskal–Wallis rank sum test, followed by Mann–Whitney test (two-sided) with Benjamini–Hochberg correction for multiple comparisons between groups of not normally distributed data.

Where representative images are shown, we observed similar results in a total of three experimental repeats of the same clones (Figs. 1c, i, 2f, g, 4e, 5b, c), two experimental repeats of two independent clones (Fig. 1f, Supplementary Fig. 1f), or two experimental repeats of the same clones (Supplementary Figs. 1g, 2d).

**Reporting summary**. Further information on research design is available in the Nature Research Reporting Summary linked to this article.

## Data availability

Source data for all main and supplementary figures are available at Mendeley Data[66]. All data are available from the authors upon reasonable request. RNA-Seq data for this submission are available at GEO as data set GSE137091. Phosphoproteomics data for this submission are available at PRIDE as data set PXD020272.

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

## Acknowledgements
We thank Adhil Mohammed, Michele Giannattasio, Gururaj Rao Kidiyoor, Giulia Bastianello, Silvia Biagini, Ortensia Franzini, Giuseppe Martano, and Angela Bachi for experimental support, helpful discussions, and help with data interpretation. We thank Barnabas Szakal for providing the tet-Rad53-Myc-AID strain. We thank the reviewers for constructive input. C.B. was supported by fellowships from Associazione Italiana per la Ricerca sul Cancro (AIRC) Fellowship i-Care (Marie Curie co-funded by the European Union, ID 16173) and the European Commission (EC-FP7-SIPOD, ID PCOFUND-GA-2012-600399). This work was supported by grants from Fondazione AIRC under IG 2015 (M.F., ID 16770), IG 2018 (M.F., ID 21416), and IG 2017 (M.P.L., ID 19783), by the Ministero dell'Istruzione/Ministero dell'Università e della Ricerca (M.F., MIUR-PRIN-15-FOIANI) and by Progetti di Ricerca di Interesse Nazionale (PRIN) 2015 (M.P.L.). E. Fabre acknowledges Labex "Who am I?" (ANR-11-LABX-0071, Idex ANR-11-IDEX-0005-02) and Cancéropôle Ile de France (ORFOCRISE PME-2015).

## Author contributions
Conceptualization: C.B. and M.F.; software: C.B.; formal analysis, C.B., M.C.L., R.B., and A.W.; investigation, C.B., A.A., E. Ferrari., M.C.L. R.B., R.C., M.G., and A.W.; data curation, C.B.; writing—original draft, C.B., E. Ferrari., and M.F.; writing—review & editing, C.B., A.A., E. Ferrari., M.C.L., S.L., M.P.L., E. Fabre, M.B.S., M.F.; visualization, C.B., M.C.L.; supervision, C.B., S.L., E. Fabre., M.B.S., and M.F.; project administration, C.B., and M.F.; funding acquisition: C.B. and M.F.

## Competing interests
The authors declare no competing interests.
