## [Peer Review File · Nature Communications]

Reviewers' comments:

Reviewer #1 (Remarks to the Author):

In this manuscript entitled: "The Rad53CHK1/CHK2-Spt21NPAT and Tel1ATM axes couple glucose tolerance to histone dosage and subtelomeric silencing", the authors investigate the links between histone homeostasis and glucose metabolism. They show that Rad53 and Spt21 regulation of histone levels mediate tolerance to glucose restriction. Additionally, the authors explore potential mechanisms including subtelomeric silencing and acetyl-CoA regulation to expand on the initial findings.

The following points should be considered when revising this manuscript:

Major:

1. Overall, the narrative of the manuscript is confusing. The results section is clearly defined in different sections that make sense and are well written independently, but are poorly linked with each other. Additionally, in the introduction, the authors jump back and forth between a few subjects without a clear argument or a direct link to the results section. For example, there is a substantial section describing DNA damage and DDR while the rest of the paper explores non-DDR functions of Rad53. Finally, the claims made throughout lines 12 onwards in the abstract are too broad and only partly supported by the data presented.
2. The authors claim in Figure 4 that hyper-acetylation of histones leads to glucose dependence by limiting Ac-CoA. While it is shown that Rad53 deletion leads to reduction of Ac-CoA, it is also necessary to show what happens to the TCA metabolites and Ac-CoA following Spt21S276A-driven histone acetylation. In line with this, figure S4A should be included as part of the main figure. Furthermore, although acetate alleviated the glucose consumption, no report of the levels of TCA cycle intermediates or Ac-CoA following the rescue with acetate were presented. Metabolomics experiments using labelled acetate should be used to show the incorporation of acetate into Ac-CoA.
3. Overall, the section on subtelomeric silencing on page 6 is confusing, and mainly correlative evidence is used to draw conclusions that are not clearly supported by data. Firstly, the authors show global mRNA alterations that are histone dependent, none of which are pursued any further. Additionally, subtelomeric derepression appears to be a general response to different cellular stress, the implications specific to glucose starvation are not controlled appropriately. In line with this, there are a few mentions of subtelomeric gene repression being independent from the carbon source, which somehow contradicts the general model presented without any further justification.

Minor:

1. In various experiments throughout the manuscript, the authors use EtOH or low glucose conditions interchangeably. It would be essential for the interpretation of those experiments to show some evidence that the metabolic changes observed are similar under both conditions.
2. The panels presenting yeast growth experiments are not clearly explained in the figure legends and are unclear to a reader with limited knowledge about yeast as a model. Additionally, quantification of those growth assays would add more robustness to the data.
3. In different experiments, 0.01% or 0.02% glucose conditions are used inconsistently without any justification.

4. The use of superscript to identify the human homolog of the genes/proteins mentioned is useful and should be used across the whole manuscript.

5. Western blots with accompanying bar chart quantification should be described as such in the figure legends.

6. Figure 2D: Which "arrows" is the figure legend referring to?

7. There are a few statements that are very generic or confusing, consider rewording those to be more relevant to the subject:

o Page 3, line 19: "Metabolic adaptation and reprogramming involve transcriptional programs"

o Page 3, line 7: "Metabolic pathways and cell division [...] metabolite pools"

o Page 3, line 31: "Reactive metabolites are a major source of endogenous DNA lesions"

8. Apart from the RNA-Seq and metabolite analysis, the statistics presented in the figures are not described. All statistical methods should be described clearly in the methods and figure legends.

Reviewer #2 (Remarks to the Author):

It has been known that histone dosage is connected to the DNA damage response through the DNA damage signaling checkpoint kinase Rad53. Here the authors extend those observations by showing that Rad53 mutants have metabolic changes that render them sensitive to glucose. They explore the reasons for the glucose sensitivity and link this to changes in histone modification and gene silencing of subtelomeric chromosome regions. Finally, the authors suggest that in cancer cells, the DDR kinases may coordinate epigenetic modifications to metabolism in a similar way and may influence carbon-directed cancer treatments.

This is an extremely complex topic and the yeast system lends itself well to deciphering the key components of the process of linking DNA repair and carbon metabolism to histone levels. The work is well done but the presentation is very dense, in both the text and the figures. Even the abstract and introduction are difficult to get through because so many points of information are presented, the direct flow of ideas gets lost. The focus seems to be on glucose tolerance and the DNA damage response, but this is not apparent in the abstract until halfway through it.

It seems that the TEL1 mutant TEL1-hy909 is key to some of the studies and really separates subtelomeric repression from histone control. This should be emphasized more and perhaps merits its own subsection in the Main results section.

Figure images of drop tests. The growth differences on limiting glucose are more apparent in the online images than the printed images. The images seem to be too contrasty.

Figure 1C. Why is the effect of limiting glucose more dramatic for H4 levels than H3 levels?

Figure 2F. What is the difference between the two red samples for 14R? Why do they give different P values compared to *sml1Δ*?

Figure S4. It is interesting that the THO/TREX complex comes up in the screen. It was reported by Schneiter et al that *acc1* is synthetic lethal with *hpr1*, perhaps linking histone levels or modifications with DNA damage and subcellular localization of damage. Similarly, Fan et al found that *hht1-hhf1* combined with *hpr1* had growth defects, again linking histone levels with DNA damage from this mutant.

Reviewer #3 (Remarks to the Author):

This work presents evidence in budding yeast that a basal level of Rad53 downregulates the histone level through the inhibition of Spt21. Upon Rad53 loss, the level of histone increases leading to glucose dependency. By the elegant use of several separation function mutants, the authors show that maintaining a physiological level of histone by Rad53-Spt21 is required for glucose tolerance via two parallel pathways: derepression of subtelomeric genes and acetyl-coA regulation by histone acetylation. These results are important, providing a new link between central DDR factors and metabolic homeostasis. The following points should be addressed in a revised version of the manuscript.

- It is not clear how the two pathways downstream of histone dosage are independent since an aberrant level of histone acetylation is expected to alleviate TPE.
- It was previously shown that an inhibition of the TOR nutrient sensing pathway counteracted TPE through a mpk1-dependant Sir3 phosphorylation. Even if the authors mention that TOR inhibition does not lead to histone increase, the authors should explore deeper the connection between the Rad53-Spt21 and the Tor-mpk1-Sir3 pathway upon glucose starvation and Tor inhibition by rapamycin.
- It is unclear from the presented data whether Rad53 activation upon DNA damage will further potentiate the spt21 pathway of histone regulation.

Reviewer #1 (Remarks to the Author):

In this manuscript entitled: “The Rad53CHK1/CHK2-Spt21NPAT and Tel1ATM axes couple glucose tolerance to histone dosage and subtelomeric silencing”, the authors investigate the links between histone homeostasis and glucose metabolism. They show that Rad53 and Spt21 regulation of histone levels mediate tolerance to glucose restriction. Additionally, the authors explore potential mechanisms including subtelomeric silencing and acetyl-CoA regulation to expand on the initial findings.

First of all, we would like to thank this Reviewer for the constructive comments. We performed various experiments to address the raised points and improved the writing style. We believe that the revised manuscript is more robust and accessible.

The following points should be considered when revising this manuscript:

Major:

1. Overall, the narrative of the manuscript is confusing. The results section is clearly defined in different sections that make sense and are well written independently, but are poorly linked with each other. Additionally, in the introduction, the authors jump back and forth between a few subjects without a clear argument or a direct link to the results section. For example, there is a substantial section describing DNA damage and DDR while the rest of the paper explores non-DDR functions of Rad53. Finally, the claims made throughout lines 12 onwards in the abstract are too broad and only partly supported by the data presented.

We improved the flow of the manuscript by removing less relevant sections from the introduction and improving the connection between results sections. We avoided generalized statements in the abstract.

2. The authors claim in Figure 4 that hyper-acetylation of histones leads to glucose dependence by limiting Ac-CoA. While it is shown that Rad53 deletion leads to reduction of Ac-CoA, it is also necessary to show what happens to the TCA metabolites and Ac-CoA following Spt21^{S276A}-driven histone acetylation. In line with this, figure S4A should be included as part of the main figure. Furthermore, although acetate alleviated the glucose consumption, no report of the levels of TCA cycle intermediates or Ac-CoA following the rescue with acetate were presented. Metabolomics experiments using labelled acetate should be used to show the incorporation of acetate into Ac-CoA.

In our original manuscript we show that *sm1Δrad53Δ* cells have lowered Ac-CoA and various central carbon metabolites, even in 2% glucose, and that this is dependent on histones (original Figure 4D-F). Further, we found that acetate, which is known to elevate histone acetylation, partially rescues the glucose dependence of *sm1Δrad53Δ* cells (original Figure 4H). We have performed the requested experiments and obtained results that strengthen the link between histone dosage, acetate and central carbon metabolites:

We acquired the metabolome profiles of *sm1Δspt21S276A*, which show elevated histone levels and histone acetylation. Indeed, our new data show that the *spt21^{S276A}* mutation phenocopies the global metabolome alterations in *sm1Δrad53Δ* cells (new Figures S6F and S6G). Importantly, Ac-CoA, downstream glycolysis intermediates and Ac-CoA-derived fatty acids are reduced in *sm1Δspt21^{S276A}* vs. *sm1Δ* cells (new Figures 4F, 4G and S6G). These data support a role of excess histones acting upstream

of the metabolic alterations. We have performed the above analysis under normal glucose conditions, because this allows us to quantify Ac-CoA. Unfortunately, the slow growth of *sm11Δspt21^{S276A}* in ethanol and in low glucose is more extreme than in *sm11Δrad53Δ* cells (see original Figure 1H), and we therefore cannot use the model for metabolite measurements under glucose restriction. The main pathway that is affected by *RAD53* deletion under glucose restriction but not in normal glucose is the TCA cycle. Since we cannot draw conclusions on TCA intermediates under glucose restriction in the *sm11Δspt21^{S276A}* model for technical limitations, we tuned down statements on the TCA cycle when describing alterations in the corresponding results section.

To address the effect of acetate on metabolites under glucose restriction, we acquired the metabolome of *sm11Δrad53Δ* cells cultured in low glucose with and without acetate pulse. Our new analysis shows that acetate increases the amount of various fatty acids, sterol precursors and TCA intermediates Citrate, Aconitate and Succinate (new Figure 4H). However, our new analysis shows that the TCA intermediates elevated by acetate are not the same TCA intermediates that are consistently lowered in *sm11Δrad53Δ* vs. *sm11Δ* cells in ethanol or low glucose (alpha-Ketoglutarate, Fumarate, Malate). Our new data thus suggest that several Ac-CoA-derived metabolites are replenished by acetate, but that the phenotypic rescue achieved by acetate does not correlate with the replenishment of relevant TCA intermediates.

We have initiated experiments on Ac-CoA detection after ¹³C acetate pulse labeling. However, we have faced technical issues to quantify Ac-CoA under low glucose conditions, and attempts to optimize the analysis are blocked by the COVID-19 measures which affect our collaborator (Laxman lab). However, the rapid elevation of fatty acid and sterol precursor levels after an acetate pulse supports the idea that acetate supply can increase Ac-CoA-derived metabolites under glucose restriction, and therefore indirectly supports the use of acetate for Ac-CoA synthesis. It is known that the direct synthesis of Ac-CoA from environmental acetate is an essential process for cells grown on acetate (De Virgilio *et al.*, 1992). Direct rapid Ac-CoA synthesis from ¹³C acetate has been reported by Cai *et al.*, 2011 (DOI 10.1016/j.molcel.2011.05.004). Although we do not know the relative contribution of low glucose and acetate to specific metabolites, our data together with the above literature suggest that externally supplied acetate can contribute to intracellular Ac-CoA-derived metabolites.

3. Overall, the section on subtelomeric silencing on page 6 is confusing, and mainly correlative evidence is used to draw conclusions that are not clearly supported by data. Firstly, the authors show global mRNA alterations that are histone dependent, none of which are pursued any further. Additionally, subtelomeric derepression appears to be a general response to different cellular stress, the implications specific to glucose starvation are not controlled appropriately. In line with this, there are a few mentions of subtelomeric gene repression being independent from the carbon source, which somehow contradicts the general model presented without any further justification.

This comment includes 3 major points.

1) “the authors show global mRNA alterations that are histone dependent, none of which are pursued any further”

The main conclusions from the RNA-Seq experiments are 1) the histone- and SIR-dependent subtelomeric gene repression in *sm11Δrad53Δ* cells and 2) the otherwise functional global response to the carbon source switch. Dissecting possible contributions of global mRNA alterations is not the scope

of this study. Therefore, we removed the original Figure 2B, restructured the Figure 2 to put clear emphasis on the subtelomere phenotype and modified the Results section accordingly.

2) “Additionally, subtelomeric derepression appears to be a general response to different cellular stress, the implications specific to glucose starvation are not controlled appropriately. In line with this, there are a few mentions of subtelomeric gene repression being independent from the carbon source, which somehow contradicts the general model presented without any further justification.”

While we show that the de-repression of subtelomeric genes is stimulated by carbon source switch (original Figure 2B), we also show that subtelomeric genes are more repressed in *sml1Δrad53Δ* cells both in 2% glucose and in ethanol (original Figure 2D). Thus, histone-mediated repression is a general, carbon source-independent phenomenon, which reduces the full inducibility of carbon source response genes within subtelomeres. As pointed out, this implies that other subtelomeric stress responses may similarly be impaired by subtelomere repression in *sml1Δrad53Δ* cells. We therefore tested rapamycin (inhibitor of TOR) and chlorpromazine (inducer of membrane stress) as other stresses that induce subtelomeric de-repression by activation of the MAP kinase Mpk1 (Ai, Bertram et al. 2002). Indeed, *sml1Δrad53Δ* cells were sensitive to rapamycin in a *SIR2*-dependent manner (new Figure S2G). Consistently, *TEL1-hy909* cells were also sensitive to rapamycin, and their sensitivity was completely rescued by *SIR2* deletion (new Figure S5C). While, *sml1Δrad53Δ* cells were also sensitive to chlorpromazine, this sensitivity was not suppressed by *SIR2* deletion (new Figure S2G), suggesting that SIR-mediated subtelomere repression in *sml1Δrad53Δ* cells contributed to rapamycin but not chlorpromazine sensitivity. In summary, subtelomere repression in *sml1Δrad53Δ* cells is a constitutive phenomenon that can impair stress responses that require subtelomeric de-repression (carbon source switch, rapamycin). Together, this suggests that a general, SIR complex-mediated subtelomere repression in *sml1Δrad53Δ* cells blunts subtelomeric stress responses, including the response to glucose starvation, resulting in a glucose dependence phenotype.

To strengthen this conclusion, we ruled out a specific role of Rad53 in the Mpk1-Sir3 pathway, which mediates subtelomere de-repression after rapamycin (Ai, Bertram et al. 2002) (new Figure S2H). We further demonstrated that carbon source switch does not de-repress subtelomeres through the Mpk1-Sir3 pathway (new Figure 2K). The independence of rapamycin and carbon source switch further support a general role of Rad53 in subtelomere expression, and not a specific role in either of the two stress response signaling pathways.

In the revised manuscript, we clearly describe the concept of general subtelomere repression (“*We next asked if Rad53 affected the expression of subtelomeric switch response genes in general, or specifically their induction in response to glucose starvation...*”). We also introduced a new Results section on rapamycin sensitivity and Mpk1-Sir3 pathway analysis. Further, we changed the term “Subtelomere switch” in our model (Figure 5) to “Subtelomere expression”, which does not imply a specific starvation signaling defect but reflects a general defect in subtelomere expression in *sml1Δrad53Δ* cells.

3) “mainly correlative evidence is used to draw conclusions that are not clearly supported by data”

Our original data show genetically that subtelomere silencing mediates glucose dependence (original Figure 2H), and that the silencing is dependent on excess histones (original Figure 4E) and mediated by the SIR complex (original Figure 2G). In the revised manuscript we have added new data connecting excess histones with the SIR complex and linking subtelomere repression to glucose dependence.

It has been known that subtelomeric gene silencing by the SIR complex is enhanced by elevated Sir3 expression levels. We have now found that *sml1Δrad53Δ* cells contain higher levels of the Sir3 protein and mRNA, and this is dependent on excess histones. This suggests that subtelomere hyper-silencing in *sml1Δrad53Δ* cells is at least partially mediated through Sir3 levels (new Figure 2L). We added a discussion point that excess histones may stabilize Sir3 protein (“*Since Sir3 levels determine subtelomeric silencing...*”).

We have analyzed cellular processes induced under glucose restriction in *sml1Δrad53Δ* cells to show which process is defective in *sml1Δrad53Δ* cells and restored by subtelomeric derepression. We show that autophagy (new Figure S3A) and mitochondrial proteins (new Figure S3B) are similarly induced by ethanol in *sml1Δrad53Δ* and *sml1Δ* cells and do therefore not account for the glucose dependence of *sml1Δrad53Δ* cells. We quantified ATP levels as measure for energy charge but found that ATP levels per cell volume were slightly elevated in *sml1Δrad53Δ* cells in comparison with *sml1Δ* cells, suggesting that a reduced energy charge cannot account for the glucose dependence of *sml1Δrad53Δ* cells (new Figure S3C). Consistently, we found similar oxygen consumption rates in *sml1Δ* and *sml1Δrad53Δ*, indicating normal respiration rate (new Figure for Reviewers 1). Since subtelomeric genes are involved in amino acid uptake and metabolization, we also quantified protein synthesis rate in *sml1Δrad53Δ* and *sml1Δ* cells by incorporation of a Methionine analog into nascent proteins. However, also the protein synthesis rate was unchanged in *sml1Δrad53Δ* cells in comparison with *sml1Δ* cells (new Figure S3D). Since subtelomeres harbor a large group of genes involved in redox processes, glucose restriction increases oxidative metabolism, and DDR mutants are sensitive to endogenous oxidative stress, we measured levels of endogenous redox-sensitive metabolites in *sml1Δ*, *sml1Δrad53Δ* and *sml1Δrad53Δsir2Δ* cells under glucose restriction by mass spectrometry. We found that *sml1Δrad53Δ* cells had a lower NADH/NAD⁺ ratio than *sml1Δ* cells (new Figure 2I), indicative of a more oxidative environment. This was associated with lower levels of the endogenous anti-oxidant N-acetylcysteine (NAC) and oxidized glutathione (new Figure 2I), implying less endogenous anti-oxidant potential. Importantly, *SIR2* deletion partially reverted these metabolite alterations (new Figure 2I). Exogenously applied NAC alleviated the slow growth of *sml1Δrad53Δ* cells under glucose restriction (new Figure 2J), suggesting that reduced anti-oxidant potential in *sml1Δrad53Δ* cells contributed to glucose dependence. The growth benefit by NAC was larger in *sml1Δrad53Δ* than in *sml1Δsir2Δrad53Δ* cells (new Figure 2J), supporting the idea that subtelomere expression provided tolerance to glucose restriction at least in part through enhancing anti-oxidant potential.

Figure for Reviewers 1. Oxygen consumption rates were measured in cells of the indicated genotypes after 20h adaptation to YP + 3% ethanol. EtOH = ethanol

To identify subtelomeric genetic mediators of glucose dependence, we performed an analysis of glucose dependence of all subtelomeric gene deletion mutants contained in the viable haploid synthetic genetic array screening library (102 genes, new Figure S4A, B). This analysis identified two glucose-dependent

mutants (*ymr315wΔ*, *yol162wΔ*). However, both genes were not strongly ethanol-inducible, not repressed in *sml1Δrad53Δ* cells and their expression was not increased by *SIR2* deletion (new Figure S4C). As a complementary approach, we manually deleted the 12 repressed gene clusters depicted in new Figure S2A, covering 57 genes in total. However, none of the deletions caused slow growth or glucose dependence (new Figure S4D). Most subtelomeric genes appear in several copies (PAU family, COS family, telomeric repeat helicases, alcohol dehydrogenases and many others) distributed over several different subtelomeres. Our efforts suggest that the simultaneous repression of several subtelomeres may be necessary to cause glucose dependence, whereas single gene deletions can be compensated for.

Since gene deletions could not identify functional contributors of glucose dependence, we conducted an overexpression screen of individual subtelomeric genes. We selected the subtelomeric genes contained in the GST collection (horizon Cat. #YSC4423), the largest genome-wide over-expression resource for yeast. We screened for suppressors of glucose dependence using an *sml1Δrad53Δ* query strain. We included the *RAD53* construct from the library as positive suppressor control. We screened on YP + 3% ethanol medium and found efficient rescue by the *RAD53* plasmid, as expected (new Figure for Reviewers 2). However, none of the subtelomeric over-expressions could efficiently rescue the growth of *sml1Δrad53Δ* cells. The largest potential suppressions that we obtained in this screen were colony size increases of 10% (new Figure for Reviewers 2), which were not visible in a spot assay setting. We obtained similar results on low galactose media, which mimic the low glucose conditions that we used throughout the study while inducing high expression of the library constructs. Glucose represses the expression of the library constructs, precluding the use of low glucose media for the screen. In summary, our genetic screening approaches suggest that a coordinate de-repression of several subtelomeric genes may be necessary to convey tolerance to glucose restriction, although the available tools did not allow us to identify the individual genetic contributors. We added a paragraph on the genetic screens in the results section (“To identify the subtelomeric genetic mediators of glucose dependence, we screened...”).

Figure for Reviewers 2. Screening a subtelomeric gene overexpression library for suppressors of glucose dependence. An *sml1Δrad53Δ* query strain was crossed into the library, *sml1Δrad53Δ* segregants overexpressing single subtelomeric genes were selected and suppression of glucose dependence was scored by colony sizes on YPE (YP + 3 % ethanol) medium. Each dot represents the mean colony size of a library gene, corrected for the standard deviation withing colonies of the respective library gene, and normalized to the median of all colonies. The y axis shows the log-transformed normalized colony sizes with values larger than 0 indicating sizes larger than the median colony size. The library genes are sorted along the x axis according to the mean colony size. The *RAD53* library construct was used as positive suppressor control.

In summary, our new data show a link between excess histones and the SIR complex, point out oxidative stress resistance as link between subtelomere expression and glucose starvation tolerance and suggest that coordinate expression of multiple genes likely mediates glucose starvation tolerance.

Minor:

1. In various experiments throughout the manuscript, the authors use EtOH or low glucose conditions interchangeably. It would be essential for the interpretation of those experiments to show some evidence that the metabolic changes observed are similar under both conditions.

We acquired global metabolome profiles of *sml1Δ* and *sml1Δrad53Δ* cells in low glucose. We added a central carbon pathway summary of the alterations caused by both glucose deprivation methods in *sml1Δ* cells, which shows overall high correlation and similarity of glucose deprivation-induced changes (new Figure S7A, B). We added data on fatty acids, sterol precursors and TCA intermediates into the main Figures (new Figure 4H). This analysis allowed us to conclude that the alterations in fatty acids and sterol precursors in *sml1Δrad53Δ* vs *sml1Δ* cells are consistent in both conditions. It further allowed us to identify the specific TCA intermediates that are consistently altered in *sml1Δrad53Δ* vs *sml1Δ* cells. Together with the acetate pulse data described above, this allowed us to conclude that alterations in TCA intermediates do not strictly correlate with the glucose dependence phenotype. We added the new finding in the Results text.

2. The panels presenting yeast growth experiments are not clearly explained in the figure legends and are unclear to a reader with limited knowledge about yeast as a model. Additionally, quantification of those growth assays would add more robustness to the data.

We improved the experiment explanations in the Figure legend and mentioned the name of the assay (semi-quantitative spot assay) at the first appearance in the results section. Please note that this assay is widely used in yeast genetic studies. The spot assay is a semi-quantitative readout and quantification of this assay is not common. By the choice of serial dilution range (in this case 6-fold dilution increase in each column) it is achieved that a visual difference represents a meaningful difference in growth rate. We added quantitative liquid growth experiments for genetic interactions between *SIR2* and histone point mutants (Figure S5B) which support the original claim.

3. In different experiments, 0.01% or 0.02% glucose conditions are used inconsistently without any justification.

The low glucose sensitivity phenotype of *sml1Δrad53Δ* cells is evident from 0.01% to 0.05% glucose as shown in original Figure S1A. We generally used 0.02% - 0.04% glucose in liquid culture to avoid growth stagnation during the experiment while obtaining higher numbers of cells for analysis. A detailed explanation for the choice of glucose concentrations is now provided in the Materials and Methods section of the revised manuscript.

4. The use of superscript to identify the human homolog of the genes/proteins mentioned is useful and should be used across the whole manuscript.

In the original manuscript we included the superscript for Mec1, Tel1, Rad53, Chk1 and Sir2.

In the revised manuscript we added the superscript for Rpd3^{HDAC1/2/3/8}, and Mpk1^{ERK5}. Although Dun1 has a high homology to human CHK2, it is functionally not directly comparable. We therefore avoid this comparison.

The histone loci were described at the first appearance but the high number of histone genes in mammalian cells does not allow a definite assignment.

For the following genes, definite mammalian sequence or functional homologs are not clearly defined, and we have therefore not included the superscript: Sml1, Spt10, Cos proteins, Rmr1, Sir1, Sir3, Sir4, Ost4, Git1 (homology to various transporters but no known functional homolog), *Ymr315w* and *Yol162w*. For the mentioned yeast transcription factors, various human homologs exist and it is not possible to denominate a single functional homolog (Mig1, Nrg1, Msn2/4).

5. Western blots with accompanying bar chart quantification should be described as such in the figure legends.

We added a clear indication in the Figure legends in the revised manuscript.

6. Figure 2D: Which “arrows” is the figure legend referring to?

The reference to arrows was wrong and has been removed from the revised manuscript.

7. There are a few statements that are very generic or confusing, consider rewording those to be more relevant to the subject:

o Page 3, line 19: “Metabolic adaption and reprogramming involve transcriptional programs”

We removed “Metabolic pathways and” to put the emphasis on the regulation of cell division by metabolic state.

o Page 3, line 7: “Metabolic pathways and cell division [...] metabolite pools”

The paragraph has been restructured and the statement removed.

o Page 3, line 31: “Reactive metabolites are a major source of endogenous DNA lesions”

The paragraph has been restructured and the statement removed.

8. Apart from the RNA-Seq and metabolite analysis, the statistics presented in the figures are not described. All statistical methods should be described clearly in the methods and figure legends.

We added a description of statistical methods in the Methods and Figure Legends. We also added a spreadsheet in the raw data with the detailed output of all statistical comparisons.

Reviewer #2 (Remarks to the Author):

It has been known that histone dosage is connected to the DNA damage response through the DNA damage signaling checkpoint kinase Rad53. Here the authors extend those observations by showing that Rad53 mutants have metabolic changes that render them sensitive to glucose. They explore the reasons for the glucose sensitivity and link this to changes in histone modification and gene silencing of subtelomeric chromosome regions. Finally, the authors suggest that in cancer cells, the DDR kinases may coordinate epigenetic modifications to metabolism in a similar way and may influence carbon-directed cancer treatments.

We would like to thank this Reviewer for his very helpful comments. We feel that the comments helped particularly to improve structure and Figure quality of the manuscript and make it thus more accessible and appealing to the readership.

This is an extremely complex topic and the yeast system lends itself well to deciphering the key components of the process of linking DNA repair and carbon metabolism to histone levels. The work is well done but the presentation is very dense, in both the text and the figures. Even the abstract and introduction are difficult to get through because so many points of information are presented, the direct flow of ideas gets lost. The focus seems to be on glucose tolerance and the DNA damage response, but this is not apparent in the abstract until halfway through it.

We improved the flow of the manuscript by restructuring the introduction, removing less relevant sections from the introduction and improving the connection between results sections. We avoided generalized statements in the abstract.

It seems that the TEL1 mutant TEL1-hy909 is key to some of the studies and really separates subtelomeric repression from histone control. This should be emphasized more and perhaps merits its own subsection in the Main results section.

We would like to thank the Reviewer for raising this point. We agree that the *TEL1-hy909* allele is the clearest model to separate the effects of subtelomere silencing and histone dosage on glucose dependence. As suggested, we have put more emphasis on the *TEL1-hy909* model by introducing a new subsection in the results and by including the *TEL1-hy909* data in the abstract. In the revised manuscript, we have also used the *TEL1-hy909* allele to test the impact of subtelomere hyper-silencing on other stresses that induce the expression of subtelomeric genes. We show that the *TEL1-hy909* allele causes *SIR2*-dependent rapamycin sensitivity (new Figure S5C).

Figure images of drop tests. The growth differences on limiting glucose are more apparent in the online images than the printed images. The images seem to be too contrasty.

We have reduced the contrast of all drop test images, which improved the quality of the paper in print. In addition to the contrast, the image quality was compromised by the import into a combined manuscript word file, which will be avoided in the final manuscript version.

Figure 1C. Why is the effect of limiting glucose more dramatic for H4 levels than H3 levels?

We have observed this effect consistently across experiments but have not commented on it in the manuscript. We do not see this difference in the histone H3 and H4 mRNA levels (original Figure S1E), suggesting that either translation or stability may be higher for H4 than for H3. We have tested by individual deletion of either *HHT2* or *HHF2* which histone contributes more to the glucose dependence phenotype of *sml1Δrad53Δ* cells. We found that deletion of *HHF2* only marginally rescues the glucose dependence of *sml1Δrad53Δ* cells, whereas deletion of *HHT2* strongly rescues the glucose dependence (new Figure for Reviewers 3). Hence, although the fold-change is higher for histone H4 at protein level, our data suggest that H3 accumulation contributes more to the glucose dependence phenotype. For this reason we decided to not pursue the stronger H4 accumulation as it is not relevant to explain the glucose dependence phenotype. However, we have specifically added this consideration to the Discussion section (“While we consistently observe stronger protein level imbalances of histone H4 than H3...”).

Figure for Reviewers 3. Spot assay of histone deletion mutants. 10^7 cells / mL were serially diluted (1:6), spotted on YP plates with the indicated carbon sources and grown for 2d. D = glucose, DR = glucose restriction (0.01% in solid media)

Figure 2F. What is the difference between the two red samples for 14R? Why do they give different P values compared to *sml1Δ*?

The two red sample are two independent *sml1Δrad53Δ* clones. The obtained result is significant for subtelomere 5R for both clones. For subtelomere 14R, clone 1 shows a trend and clone 2 shows significance. We now clearly labeled the different clones in the Figure (new Figure 2G), and added a better description in the Figure Legends and Results text.

Figure S4. It is interesting that the THO/TREX complex comes up in the screen. It was reported by Schneiter et al that *acc1* is synthetic lethal with *hpr1*, perhaps linking histone levels or modifications with

DNA damage and subcellular localization of damage. Similarly, Fan et al found that *hht1-hhf1* combined with *hpr1* had growth defects, again linking histone levels with DNA damage from this mutant.

We find this connection very interesting. Our previous study has shown that deletion of *HPR1* rescues HU sensitivity of DNA damage response mutants (Bermejo, Capra et al. 2011). The similarity between *sm11Δrad53Δ* and *hpr1Δ* cells stems mainly from repressed subtelomeric genes. It would be interesting in the future to test the implication of this silencing in *hpr1Δ* cells, keeping in mind that histone gene deletions have opposite effects on growth speed in *hpr1Δ* and *sm11Δrad53Δ* cells. Following this comment, we have tested the effect of partial AID-mediated degradation of the essential *Acc1* protein and of the low expression *ACC1* DAmP allele in *sm11Δrad53Δ* cells. However, we do not find a similar negative interaction here, suggesting that the relationship may be more complex (data not included). While the connection between the DNA damage response and the THO/TREX complex in histone regulation is a very interesting topic for a future study, we believe that discussing this topic inside the manuscript would add to the high density of the work.

Reviewer #3 (Remarks to the Author):

This work presents evidence in budding yeast that a basal level of Rad53 downregulates the histone level through the inhibition of Spt21. Upon Rad53 loss, the level of histone increases leading to glucose dependency. By the elegant use of several separation function mutants, the authors show that maintaining a physiological level of histone by Rad53-Spt21 is required for glucose tolerance via two parallel pathways: derepression of subtelomeric genes and acetyl-coA regulation by histone acetylation. These results are important, providing a new link between central DDR factors and metabolic homeostasis.

We would like to thank this Reviewer for his helpful and constructive comments. The comments helped to clarify our claims on the link between histone acetylation and subtelomere expression in glucose starvation tolerance. We would like to specifically thank for the comment on Rad53-Spt21 and Mpk1-Sir3 crosstalk because we feel that the revised manuscript is now much better integrated with the existing literature on the topic of subtelomere expression.

The following points should be addressed in a revised version of the manuscript.

- It is not clear how the two pathways downstream of histone dosage are independent since an aberrant level of histone acetylation is expected to alleviate TPE.

We agree that histone acetylation regulates subtelomere repression and de-repression. This is widely accepted and we do not intend to challenge the concept. The de-acetylation of H4K16 by Sir2 mediates the spreading of silent chromatin by the SIR complex. In addition, many histone acetyltransferases have been implicated in subtelomeric repression and de-repression, including Gcn5 of the SAGA and ADA complexes, the NuA3 complex, the NuA4 complex, the SAS complex, Hat1 and Rtt109 (see (Zhou, Zhou et al. 2009; Power, Jeffery et al. 2011) and many others). Similarly, hyper-acetylation of specific sites by HDAC deletion (*rpc3Δ*) is sufficient to induce excessive subtelomere silencing (Zhou, Zhou et al. 2009).

We intend to make the point that the effects of relieving subtelomeric silencing (*sir2Δ*) and abrogating N-terminal H3 acetylation (*hht2* point mutants) on glucose dependence in *sml1Δrad53Δ* cells are “separable”. The strongest evidence that histone hyper-acetylation causes glucose dependence in the absence of subtelomere silencing is the residual glucose dependence of *sml1Δrad53Δsir2Δ* cells and its rescue by abrogation of N-terminal H3 acetylation (original Figure 3A). To strengthen this argument, we have added data in the revised manuscript on the combined effect of N-terminal H3 acetylation mutants and *SIR2* deletion on the expression of SIR-repressed genes (*COS1*, *COS8*) in subtelomeres (new Figure 3B). We confirm that deletion of *SIR2* de-represses the subtelomeric *COS1* and *COS8* genes in *sml1Δrad53Δ* cells. However, the N-terminal H3 acetylation mutants do not de-repress either of the two genes in *sml1Δ* cells, and only the *hht2-5KR* but not the *hht2-5KQ* mutant increases the de-repression in *sml1Δrad53Δ* cells. Thus, our new data suggest that hyper-acetylation of the selected 5 sites (mimicked by *hht2-5KQ*) would not cause de-repression of the analyzed SIR-repressed genes. This supports the idea that SIR-mediated gene repression and global acetylation are genetically separable. We included these data in the Results section (“Although the contributions of *SIR2* and H3 acetylation sites to glucose dependence are genetically separable...”). However, given the various regulatory roles of H3 and H4 acetylation on subtelomere expression, we assume that histone hyper-acetylation likely influences subtelomere expression in some loci and thus have an additional indirect impact on glucose dependence. We added this consideration to the Discussion section (“We have shown that histone hyper-acetylation

and subtelomere silencing...). We have also weakened the expression “independent” to “separable” throughout the manuscript.

- It was previously shown that an inhibition of the TOR nutrient sensing pathway counteracted TPE through a mpk1-dependant Sir3 phosphorylation. Even if the authors mention that TOR inhibition does not lead to histone increase, the authors should explore deeper the connection between the Razd53-Spt21 and the Tor-mpk1-Sir3 pathway upon glucose starvation and Tor inhibition by rapamycin.

The final output of the TOR-Mpk1-Sir3 pathway is the phosphorylation of Sir3 which results in subtelomere de-repression after TOR inhibition, e.g. by rapamycin (Ai, Bertram et al. 2002). We tested if Myc-tagged Sir3 was phosphorylated in response to glucose deprivation (glucose-to-ethanol switch), using bandshift analysis with rapamycin as positive control. Our new data show that rapamycin but not glucose deprivation induces the phosphorylation of Sir3 (new Figure 2K).

We also measured rapamycin-induced bandshift in *sml1Δ*, *sml1Δrad53Δ*, and *sml1Δmpk1Δ* cells and found that deletion of MPK1 but not deletion of RAD53 interferes with rapamycin-induced Sir3 phosphorylation (new Figure S2H).

Both datasets together suggest that rapamycin- and ethanol-induced subtelomere de-repressions act through different mechanisms.

During these experiments we found that Sir3-Myc levels were consistently elevated in *sml1Δrad53Δ* cells in comparison with *sml1Δ* cells. We therefore obtained *sml1Δrad53Δhht2Δ* cells with Myc-tagged Sir3 by tetrad dissection and could show that the increase in Sir3 is HHT2-dependent (new Figure 2L). Our RNA-Seq data show that the *SIR3* mRNA levels are equally altered in the above genotypes as the protein levels (original Table S2). Thus, although the Mpk1 pathway seems unaffected by excess histones, our data suggest that excess histones may enhance silencing in part through transcriptional up-regulation of Sir3. We added this point to the discussion (“*Since Sir3 levels determine subtelomeric silencing...*”).

We believe that this comment was particularly helpful to integrate the subtelomeric silencing phenotype of our study into the existing literature and thereby improved the manuscript.

- It is unclear from the presented data whether Rad53 activation upon DNA damage will further potentiate the spt21 pathway of histone regulation.

We agree that the regulation of Spt21 may be stimulated by DNA damage. Indeed, we have first detected Spt21 phosphorylation on S276 after a mild treatment with MMS (Bastos de Oliveira, Kim et al. 2015). Moreover, *spt21Δ* clearly rescues the hydroxyurea sensitivity of *sml1Δrad53Δ* cells which suggests that Spt21 is inactivated in response to DNA damage to lower histone dosage (new Figure for Reviewers 4A). Consistently, the *spt21S276A* mutant is sensitive to high doses of hydroxyurea (new Figure for Reviewers 4B). However, the *hht2Δ* (encoding H3) and *hhf2Δ* (encoding H4) mutations are not sufficient to efficiently alleviate the pronounced hydroxyurea sensitivity of *sml1Δrad53Δ* cells (new Figure for Reviewers 4A). Thus, although it is possible that the Spt21 pathway regulates histones after DNA damage, H3 and H4 are not the major mediators of such regulation. This is in strong contrast to the major role of H3 regulation under glucose restriction (new Figure for Reviewers 4A). The scope of this study is the role of Rad53 in glucose restriction tolerance, and the relevant Spt21 targets are different for

the hydroxyurea response and glucose starvation. Therefore, we would prefer to avoid details on Spt21 regulation during DNA damage to keep the already complex manuscript more focused on the glucose phenotype. Therefore, we added the above observations as comment in the Discussion ("Notably, we observed that *sml1Δspt21^{S276A}* cells are sensitive...").

Figure for Reviewers 4. Spot assay of *spt21* and histone deletion mutants in hydroxyurea. 10^7 cells / mL were serially diluted (1:6), spotted on YP plates with the indicated carbon sources and drugs and grown for 2d. D = glucose, DR = glucose restriction (0.01% in solid media), HU = hydroxyurea

References

- Ai, W., P. G. Bertram, et al. (2002). "Regulation of subtelomeric silencing during stress response." *Mol Cell* **10**(6): 1295-1305.
- Bastos de Oliveira, F. M., D. Kim, et al. (2015). "Phosphoproteomics reveals distinct modes of Mec1/ATR signaling during DNA replication." *Mol Cell* **57**(6): 1124-1132.
- Bermejo, R., T. Capra, et al. (2011). "The replication checkpoint protects fork stability by releasing transcribed genes from nuclear pores." *Cell* **146**(2): 233-246.
- Power, P., D. Jeffery, et al. (2011). "Sub-telomeric core X and Y' elements in *S. cerevisiae* suppress extreme variations in gene silencing." *PLoS One* **6**(3): e17523.
- Zhou, J., B. O. Zhou, et al. (2009). "Histone deacetylase Rpd3 antagonizes Sir2-dependent silent chromatin propagation." *Nucleic Acids Res* **37**(11): 3699-3713.

REVIEWERS' COMMENTS:

Reviewer #1 (Remarks to the Author):

All my major concerns were thoroughly addressed and the text has been revised to remove overly broad statements

Reviewer #2 (Remarks to the Author):

The authors have performed several additional experiments to clarify the links between glucose metabolism, histone levels, Rad53 checkpoint function, and metabolites in the presence of acetate. The authors have also rewritten much of the text. While the manuscript remains complex and dense, the message is much clearer now and represents new insights into metabolic alterations, gene silencing and glucose metabolism and the role of Rad53/CHK1 in regulating this.

Reviewer #3 (Remarks to the Author):

The revised version adequately addressed my specific points.

Eric Gilson